



# Potential of future stratospheric ozone loss in the mid-latitudes under climate change and sulfate geoengineering

Sabine Robrecht[1], Bärbel Vogel[1], Simone Tilmes[2], and Rolf Müller[1]

[1]Forschungszentrum Jülich, Institute for Energy and Climate research - Stratosphere (IEK-7), Germany
[2]National Center for Atmospheric Research, Atmospheric Chemistry Observations and Modeling Lab, Boulder, CO, USA

*Correspondence to:* Sabine Robrecht (sa.robrecht@fz-juelich.de)

**Abstract.** The potential of heterogeneous chlorine activation in the mid-latitude lowermost stratosphere during summer is a matter of debate. The occurrence of heterogeneous chlorine activation through the presence of aerosol particles could cause ozone destruction. This chemical process requires low temperatures and is accelerated by an enhancement of the stratospheric water vapour and sulfate amount. In particular, the conditions present in the lowermost stratosphere during the North American Summer Monsoon season (NAM) are expected to be cold and moist enough for causing the occurrence of heterogeneous chlorine activation. Furthermore, the temperatures, the water vapour mixing ratio and the sulfate aerosol abundance are affected by future climate change and by the potential application of sulfate geoengineering. Hence, both future scenarios could promote this ozone destruction process.

We investigate the likelihood for the occurrence of heterogeneous chlorine activation and its impact on ozone in the lowermost stratospheric mixing layer between tropospheric and stratospheric air above central North America (30.6–49.6°N, 72.25–124.75°W) in summer for conditions today, at the mid and at the end of the 21st century. Therefore, the results of the Geoengineering Large Ensemble Simulations (GLENS) for the lowermost stratospheric mixing layer between tropospheric and stratospheric air are considered together with 10 day box-model simulations performed with the Chemical Lagrangian Model of the Stratosphere (CLaMS). In GLENS two future scenarios are simulated: the RCP8.5 climate change scenario and a geoengineering scenario, where sulfur is additionally injected in the stratosphere to keep the global mean surface temperature from changing.

In the GLENS simulations, the mixing layer will warm and moisten in both future scenarios with a larger effect in the geoengineering scenario. The likelihood for chlorine activation to occur in the mixing layer is highest in the years 2040–2050 if geoengineering is applied, accounting for 3.3%. In comparison, the likelihood for conditions today is 1.0%. At the end of the 21st century, the likelihood of this ozone destruction process to occur decreases. We found that 0.1% of the ozone mixing ratios in the mixing layer above central North America is destroyed for conditions today. A maximum ozone destruction of 0.3% in the mixing layer occurs in the years 2040–2050 if geoengineering is applied. Comparing the southernmost latitude band (30–35°N) and the northernmost latitude band (44–49°N) of the considered region, we found a higher likelihood for the occurrence of heterogeneous chlorine activation in the southernmost latitude band, causing a higher impact on ozone as well. However, the ozone loss process is found to have a minor impact on the mid-latitude ozone column with not more than 0.1 DU today or in the future scenarios.



# 1 Introduction

Climate change and a possible application of sulfate geoengineering will affect the temperature and the composition of the air in the mid-latitude lowermost stratosphere. Especially, for the case of geoengineering using stratospheric sulfate aerosols, the potential occurrence of heterogeneous chlorine activation in the mid-latitude lowermost stratosphere in summer, which would

cause a catalytic ozone destruction, has been discussed in previous studies (Anderson et al., 2012, 2017; Clapp and Anderson, 2019; Schwartz et al., 2013; Robrecht et al., 2019; Schoeberl et al., 2020). Here, we analyse the likelihood for the occurrence of a heterogeneous chlorine activation and its impact on ozone in the lowermost stratosphere in a future climate including the hypothetical application of sulfur injections into the stratosphere.

Stratospheric ozone absorbs UV-radiation and thus protects animals, plants and also the human skin from radiative damage.

Ozone in the mid-latitude lower stratosphere between the tropopause and the 100 hPa level contributes in summer $\sim$6% (38°N) to 17%(53°N) to the ozone column (Logan, 1999). The ozone mixing ratios in the mid-latitude lower stratosphere are dominated by transport processes driven by the Brewer-Dobson-Circulation (BDC) (e.g. Ploeger et al., 2015b). However, the ozone mixing ratio in this region is additionally affected by chemical processes. The oxidation of methane and carbonmonoxide to $CO_2$ causes a production of ozone in the lowermost stratosphere (e.g. Johnston and Kinnison, 1998; Grenfell et al., 2006)

while lowermost stratospheric ozone is mainly destroyed by $HO_x$-radicals (=OH, $HO_2$, H) (e.g. Müller, 2009). In recent years, furthermore the impact of heterogeneous chlorine activation caused by an enhancement of stratospheric water vapour through convective overshooting was discussed (Anderson et al., 2012, 2017; Clapp and Anderson, 2019; Schwartz et al., 2013; Robrecht et al., 2019; Schoeberl et al., 2020).

Climate change will affect ozone abundances in the lowermost stratosphere (WMO, 2018). An increase of greenhouse gas

(GHG) concentrations is expected to cool the stratosphere (e.g. Fels et al., 1980; Iglesias-Suarez et al., 2016), slowing down gas phase ozone destruction processes. Furthermore, ozone depleting substances (ODS) will decrease in the future due to the Montreal Protocol and its amendments and adjustments (WMO, 2018). Both factors lead to an increase in upper stratospheric ozone (e.g. Haigh and Pyle, 1982; Rosenfield et al., 2002; Eyring et al., 2010; Revell et al., 2012; WMO, 2018). Since climate change would additionally lead to an acceleration of the BDC (e.g. Butchart and Scaife, 2001; Garcia and Randel, 2008;

Butchart et al., 2010; Polvani et al., 2018), more ozone could be transported from the tropics to the poles and mid-latitudes. However, an acceleration of the BDC will not be uniform throughout the stratosphere (e.g. Ploeger et al., 2015a). In addition to changes in stratospheric transport, increasing atmospheric $CH_4$ mixing ratios cause ozone formation in the lowermost stratosphere through $CH_4$ oxidation to $CO_2$, so that an increase in ozone is expected in the lowermost stratosphere for the future climate change (Iglesias-Suarez et al., 2016).

A hypothetical application of geoengineering through sulfate injections into the stratosphere aiming to cool the troposphere would likewise affect ozone abundances in the lowermost stratosphere, but in a different way than through climate change. The troposphere-to-stratosphere-transport in the mid-latitudes could be reduced due to a cooling of the troposphere and a warming of the lower stratosphere by applying geoengineering (Visioni et al., 2017b). Furthermore, the stratospheric water vapour abundance would increase, because more stratospheric sulfate particles would warm the tropical tropopause layer and





thus allow more water vapour to enter the stratosphere (Brewer, 1949; Dessler et al., 2013; Visioni et al., 2017a). An increase in stratospheric water vapour would additionally warm the stratosphere (e.g. Vogel et al., 2012; Dessler et al., 2013). Furthermore, due to a higher $H_2O$ mixing ratio, the concentration of $HO_x$-radicals increases and thus ozone destruction in the $HO_x$-cycle accelerates (Heckendorn et al., 2009; Tilmes et al., 2017). Pitari et al. (2014) describe an overall decrease

of stratospheric ozone by the mid of the 21st century when geoengineering is applied from 2020 onwards. Mid-latitude ozone is mainly affected by an increase in heterogeneous chemistry, which increases $ClO_x$ (=$Cl+ClO+2 \cdot Cl_2O_2$) and reduces $NO_x$(=$NO + NO_2 + NO_3 + 2 \cdot N_2O_5$) (Pitari et al., 2014; Heckendorn et al., 2009). The increase in $ClO_x$, which causes ozone destruction in the $ClO_x$-cycle (Stolarski and Cicerone, 1974; Rowland and Molina, 1975), is balanced by the reduction in $NO_x$, which reduces ozone destruction in the $NO_x$-cycle (Crutzen, 1970; Johnston, 1971), until the mid of this century (Pitari et al.,

2014). In the subsequent decades, the decrease of ODS would result in an general increase of stratospheric ozone (Pitari et al., 2014).

In the mid-latitude lowermost stratosphere in summer a further chemical process may affect ozone abundances (Anderson et al., 2012, 2017; Clapp and Anderson, 2019; Robrecht et al., 2019). The key step of this ozone destruction mechanism is the chlorine activation through the heterogeneous reaction

$$ClONO_2 + HCl \xrightarrow{het.} Cl_2 + HNO_3. \tag{R1}$$

Photolysis of the formed $Cl_2$ yields active chlorine radicals, which can drive catalytic ozone loss cycles based on the reactions

$$ClO + ClO + M \rightarrow ClOOCl + M, \tag{R2}$$

$$ClO + BrO \rightarrow Br + Cl + O_2 \tag{R3}$$

and $$ClO + HO_2 \rightarrow HOCl + O_2. \tag{R4}$$

These cycles are already known from polar regions, namely as the ClO-Dimer-cycle (R2, Molina and Molina, 1987) and the ClO-BrO-cycle (R3, McElroy et al., 1986). In particular a further cycle based on R4 first introduced by Solomon et al. (1986) for polar regions is expected to be relevant at activated chlorine conditions in the mid-latitude lowermost stratosphere in summer (Johnson et al., 1995; Ward and Rowley, 2016; Robrecht et al., 2019). For chlorine activation to occur, the temperature has to fall below a threshold temperature in polar regions (Drdla and Müller, 2012), which depends on the water vapour content, the

sulfate aerosol surface area density and on altitude. Robrecht et al. (2019) investigated the water vapour threshold of chlorine activation in the mid-latitude lowermost stratosphere and showed additionally a minor dependence of chlorine activation on the mixing ratio of inorganic chlorine ($Cl_y$) and nitrogen ($NO_y$).

Since low temperatures and an enhancement of water vapour above the background of 4–6 ppmv $H_2O$ are crucial for chlorine activation and thus ozone loss to occur, Anderson et al. (2012) proposed that this ozone loss mechanism is important for

the North American lowermost stratosphere in summer. There, water vapour could penetrate into the lowermost stratosphere through convective overshooting events within severe storm systems (Homeyer et al., 2014; Herman et al., 2017; Smith et al., 2017; Clapp and Anderson, 2019). How the intensity and frequency of severe storm systems will change over North America in the future, is not clear (Anderson et al., 2017). However, an increase of stratospheric sulfate particles, e.g. caused by volcanic





eruptions or the application of geoengineering, would promote heterogeneous chlorine activation and thus the occurrence of ozone destruction known from polar regions in the mid-latitude lowermost stratosphere (Anderson et al., 2012; Robrecht et al., 2019).

How likely and how wide spread this ozone loss process could occur in the future, is not yet investigated. Robrecht et al. (2019)
found that mid-latitude ozone loss through enhanced water vapour is unlikely for today's conditions analysing the chemical process and measurements of water vapour, temperature and ozone in the lowermost stratosphere. Here, we investigate the likelihood for the occurrence of this ozone loss process in the lowermost stratosphere above central North America in summer with a focus on future climate conditions. Therefore, the model results from the Geoengineering Large Ensemble Simulations (GLENS) (Tilmes et al., 2018) are analysed for the years 2010–2020, 2040–2050 and 2090–2100. In GLENS, two future sce-
narios are simulated, a climate change scenario following the representative concentration pathway 8.5 (RCP8.5) scenario and the application of sulfate geoengineering scenario, designed to keep the global mean temperature to the year 2020. In general, there are different RCP scenarios describing different pathways of radiative forcing by the year 2100. The RCP8.5 scenario assumes a worst case scenario with a high GHG emission and thus a large increase of the global mean temperature, which continues to increase after 2100 (Pachauri et al., 2014).

Based on the GLENS results, box-model simulations with the Chemical Lagrangian Model of the Stratosphere (CLaMS) (McKenna et al., 2002b, a) are initialized, which are used to calculate chlorine activation thresholds marking the threshold for chlorine activation via R1 dependent on the temperature and the water vapour mixing ratio. Hence, the chlorine activation threshold separates conditions causing and not causing chlorine activation (and thus chlorine catalysed ozone loss processes known from polar regions). Comparing the chlorine activation thresholds and the conditions in GLENS, the likelihood for chlo-
rine activation to occur is assessed and the impact of this ozone loss process on lowermost stratospheric ozone is investigated. In this paper, first the experimental setup is introduced (Sec. 2). Furthermore, the temperatures and the chemical composition of the lowermost stratosphere today and in future are analysed focussing on the GLENS results (Sec. 3). The likelihood for the occurrence of chlorine activation is determined in Sec. 4 comparing the conditions present in the GLENS results with calculated chlorine activation thresholds. An upper boundary for the impact of this ozone loss process is assessed in this study,
additionally investigating an assumption with 2 K lower temperatures. Finally, the results of this study will be discussed (Sec. 5) and summarized (Sec. 6).

## 2   Experimental setup

The GLENS results are used as a data set representing the conditions in the early (2010–2020), mid (2040–2050) and late (2090–2100) 21st century. CLaMS simulations are conducted based on the GLENS results to calculate chlorine activation
thresholds. Comparing chlorine activation thresholds calculated from CLaMS simulations and GLENS results, we assess the likelihood for ozone loss to occur in the lowermost stratosphere above central North America in summer today and in future scenarios.



## 2.1 GLENS simulations

The GLENS simulations were performed with version 1 of the Community Earth Sytem Model (CESM1, Hurrell et al., 2013). The Whole Atmosphere Community Climate Model (WACCM, Marsh et al., 2013) was used as the atmospheric component using a $0.9° \times 1.25°$ (latitude x logitude) grid and comprising 70 vertical layers up to a pressure of $10^{-6}$ hPa. WACCM is cou-
pled to land, sea ice and ocean models and includes fully interactive middle atmosphere chemistry, simplified chemistry in the troposphere as well as sulfate bearing gases important for the formation of stratospheric sulfate (Mills et al., 2017). The 3-mode version of the aerosol module (MAM3, Mills et al., 2016) was used to properly represent aerosol microphysics and the sulfate aerosol formation from injected $SO_2$.

The ability of the chosen model (CESM1 with WACCM) to properly represent both atmospheric chemistry and dynamics as
well as the atmospheric response on a severe stratospheric $SO_2$ injection was shown by Mills et al. (2017). A comparison of observations with four free running simulations for the years 1975–2016 initialized with conditions from 1 January 1975 showed a good agreement regarding temperatures, atmospheric winds, stratospheric water vapour and ozone. In particular, the model depicts the quasi biennial oscillation (QBO) and the "tape recorder" (Mills et al., 2017). Simulations of the Mt. Pinatubos eruption 1991 were in agreement with the observed aerosol optical depth. Especially, the radiative impacts (namely the
absorbed solar radiation and the outgoing long wave radiation) agreed very well with the observations which is important to properly simulate the effect of stratospheric $SO_2$ injections on stratospheric chemistry and dynamics.

An extensive overview on the GLENS simulations is given elsewhere (Tilmes et al., 2018). Briefly, GLENS simulations were performed to provide a comprehensive data set for studying the limitations, side-effects and risks of geoengineering. The GLENS study comprises three ensemble members from the year 2010 to the end of the 21st century following the RCP8.5
scenario. Since only the first of these simulations completed until 2099, we choose the first of these ensemble members for our study. We furthermore choose the first of twenty ensemble members of the geoengineering scenario comprising the years 2020–2099.

The geoengineering scenario of GLENS is based on the RCP8.5 scenario, but aims to hold the global mean temperature, the inter-hemispheric temperature gradient and the equator-to-pole gradient at the level to the year 2020 by applying stratospheric
sulfur injections (for more details see Kravitz et al. (2017)). To reach the temperature targets, $SO_2$ is simultaneously injected beginning from the year 2020 at four injection locations. These are chosen to be at $15°N$ and $15°S$ in an altitude of $25\,km$ and at $30°N$ and $30°S$ in an altitude of $22.8\,km$ based on previous studies about the injection location on the effectiveness of geoengineering (MacMartin et al., 2017; Tilmes et al., 2017). The amount of injected sulfur at each location is determined using a feedback algorithm that annually adjusts the location rates (MacMartin et al., 2014; Kravitz et al., 2016, 2017). To
reach the temperature targets, more than $50\,Tg\ SO_2$ would have been emitted at the end of the 21st century. This is five times the emitted amount of sulfur by the Mt. Pinatubo eruption in the year 1992 (Tilmes et al., 2018).





**Table 1.** Denotation of cases considered in this study.

| Case | future scenario | Years |
|---|---|---|
| C2010 | N/A | 2010–2020 |
| C2040 | Climate Change following the RCP8.5 emission pathway | 2040–2050 |
| C2090 | Climate Change following the RCP8.5 emission pathway | 2090–2100 |
| F2040 | sulfate geoengineering | 2040–2050 |
| F2090 | sulfate geoengineering | 2090–2100 |

### 2.1.1 Data selection

GLENS provides a comprehensive global data set assuming two different future scenarios (climate change and application of sulfate geoengineering) and covering the years 2010–2100. Here, only specific decades and a specific region – namely air masses in the lowermost stratosphere above central North America in the early, mid and late 21st century – are considered using the 10 day instantaneous GLENS output for the months June, July and August.

For conditions of the early 21st century, the control-run for the years 2010–2020 is used. For a future with climate change, the RCP8.5 scenario is used for the years 2040–2050 (mid of this century) and 2090-2099 (end of this century) of the same model-run. The same years are considered for the geoengineering scenario (GLENS Feedback simulations). The five considered cases in this paper are referred to as C2010 for today's conditions (2010–2020) and C2040 (2040–2050) as well as C2090 (2090–2100) for climate change scenarios. The geoengineering cases are named F2040 and F2090 for the mid and the end of the 21st century, respectively. An overview on the considered cases is given in Tab. 1.

GLENS results are selected for a latitude range of $30.6 - 49.5°N$ and a longitude range of $72.25 - 124.75°W$ (grey marked in Fig. 1, left). Since the ozone loss process focused on in this study is expected to occur most likely in summer, only the months June, July and August are considered. As shown in Fig 1 (right) the tropopause altitude varies depending on latitude and the considered case. Since the tropopause altitude varies significantly above central North America, the latitude range is divided into four bins (30–35°N, 35–40°N, 40–44°N and 44–49°N), but in this study the focus is on subtropical latitude band (30–35°) with a more likely subtropical character of the chemical composition and the extra-tropical latitude band (44–49°N) representing the chemical composition of the extra-tropics around the tropopause.

This study focuses on the mixing layer between tropospheric and stratospheric air located in the lowermost stratosphere above central North America (blue illustrated in Fig. 1, left). Without mixing between tropospheric and stratospheric air, correlations of trace gases mainly released in the troposphere (e.g. CO) and mainly produced in the stratosphere (e.g. $O_3$) form an "L-shape" (Pan et al., 2004; Vogel et al., 2011) consisting of a tropospheric and a stratospheric branch. A mixing layer between tropospheric and stratospheric air masses additionally generates mixing lines in the tracer-tracer-space resulting in "cutting off" the corner the L-shape (e.g. Hoor et al., 2002; Pan et al., 2004; Vogel et al., 2011). The mixing layer in mid-latitudes is located close the thermal tropopause, with a significant part in the lowermost stratosphere. Air masses within the mixing layer are characterised by relatively high $H_2O$ from the troposphere compared to typically low stratospheric water vapour amounts





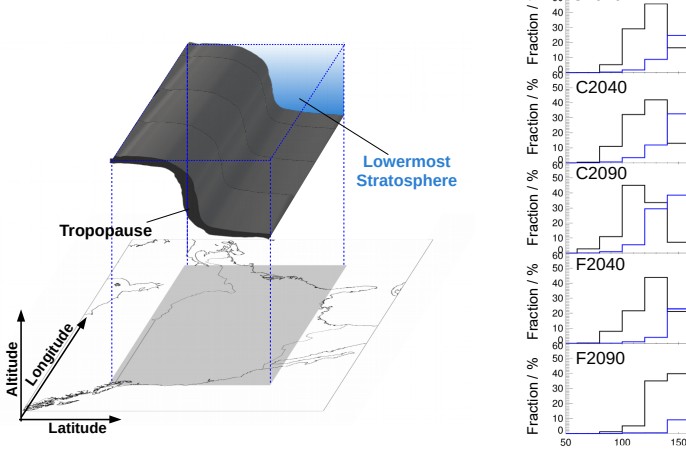

**Figure 1.** Schematic overview on the selected data region over North America (left). The position of the tropopause over central North America in summer is illustrated in black and the mixing layer is directly located above the tropopause in the lowermost stratosphere (blue). The right panel presents the tropopause pressure released from GLENS in this region depending on the latitude range and all considered cases (see Tab. 1.)

and by $O_3$ and $Cl_y$ higher than usually found in the upper troposphere from mixing with stratospheric air. Furthermore, the temperatures are low due to the location close to the thermal troposphere. Hence, the lowermost stratospheric mixing layer shows conditions for which heterogeneous chlorine activation most likely occurs and is therefore in the focus of this study.

Since the tropopause altitude and thus the altitude range of the mixing layer varies for different latitudes and future scenarios,

5 the selected altitude range for air masses in the lowermost stratospheric mixing layer is determined so that it may vary in the considered cases. The lower boundary of the data selected is chosen to be the thermal tropopause calculated according to the WMO definition within GLENS for each time step by the model. The upper boundary is determined by a mixing ratio of 35 ppbv of the artificial E90 tracer. This is a passive tropospheric tracer in WACCM globally released with a lifetime of 90 days, a mixing ratio of ∼90 ppbv at the tropopause and a strong decrease in the lowermost stratosphere (Abalos et al.,

10 2017). Since the E90-tracer is emitted continuously throughout the GLENS simulations, it is independent of possible changes in the emission rates of other tropospheric trace gases and therefore a good marker for the fraction of tropospheric air in the considered air mass.

## 2.2 CLaMS simulations

Box-model simulations with the CLaMS (e.g. McKenna et al., 2002b, a) are performed to determine chlorine activation thresh-

15 olds. CLaMS simulations are further initialized based on GLENS results. Therefore, considered GLENS results are divided in different latitude regions, pressure levels and ozone mixing ratios as shown in Tab. 2. Any combination of latitude, pressure





**Table 2.** Overview on the latitude, pressure, ozone, water vapour and temperature ranges, for which CLaMS simulations are conducted. Each combination of latitude, pressure and ozone range is summarized in a data group resulting in 100 different data groups. For a better overview in this paper, the pressure ranges are allocated to a pressure level.

| Latitude / °N | 30–35 | 35–40 | 40–44 | 44–49 | |
|---|---|---|---|---|---|
| Pressure range / hPa | 70–90 | 90–110 | 110–130 | 130–150 | 150–300 |
| Pressure level | 80 hPa | 100 hPa | 120 hPa | 140 hPa | 160 hPa |
| $O_3$ / ppbv | 150–250 | 250–350 | 350–450 | 450–550 | 550–650 |
| $H_2O$ / ppmv | 4–30 | in steps of 1 ppmv | | | |
| Temperature / K | 195–230 | in steps of 1 K | | | |

and ozone range is referred to as a data group. The pressure levels are chosen based on the vertical levels used in GLENS. The GLENS results are separated in different ozone ranges, because higher ozone mixing ratios are correlated with higher $Cl_y$ mixing ratios, which promote the likelihood for heterogeneous chlorine activation to occur and its impact on ozone (Robrecht et al., 2019). Furthermore, based on the ozone mixing ratio considered air masses can be divided in those with a more tropo-
spheric character (low ozone) and in those with a more stratospheric character (high ozone).

Stratospheric chemistry is simulated along artificial 10-day trajectories, which are designed to calculate the chlorine activation threshold for each data group. Therefore, the trajectories are located at a specific point in the stratosphere determined as 102.5°W (middle longitude over the considered longitude range) and the middle pressure and latitude of the specific data group (e.g. 32.5°N for the latitude range 30–35°N and 80 hPa for the pressure range of 70–90 hPa). As chemical initialization for the
CLaMS box-model, the median mixing ratio is taken of each trace gas from GLENS in a data group. For each of the five cases (see Tab.1) and each data group (Tab. 2), chemical simulations are conducted assuming constant water vapour varying from 4–30 ppmv in steps of 1 ppmv and a constant temperature varying from 195–230 K in 1 K steps resulting in a total of 455,000 box-model simulations. Hereafter, instead of pressure ranges the allocated pressure level as given in Tab. 2 is used in the text. Heterogeneous chemistry is only considered here to take place on liquid particles to ensure a comparability to the study of
Anderson et al. (2012). Further, only a very low fraction of GLENS data points shows conditions cold enough for the formation of ice particles. As initialization for liquid particles, the particle number density and the gas phase equivalent of $H_2SO_4$ is needed, taken from monthly GLENS data as median of a data group.

### 2.2.1 Calculation of the chlorine activation thresholds

The chlorine activation threshold for each data group defines the conditions that allow chlorine catalysed ozone destruction.
Hence, the fraction of GLENS air masses showing these conditions corresponds to the likelihood that ozone destruction occurs as a result of heterogeneous chlorine activation in the North American lowermost stratosphere.

Chlorine activation thresholds are calculated for each data group. Therefore, the water vapour and temperature conditions causing chlorine activation within a simulation are identified. Chlorine activation is assumed to have occurred, if $ClO_x$ contributes 10% of $Cl_y$ within the first 5 days of a CLaMS simulation. For each water vapour value, the maximum temperature at which





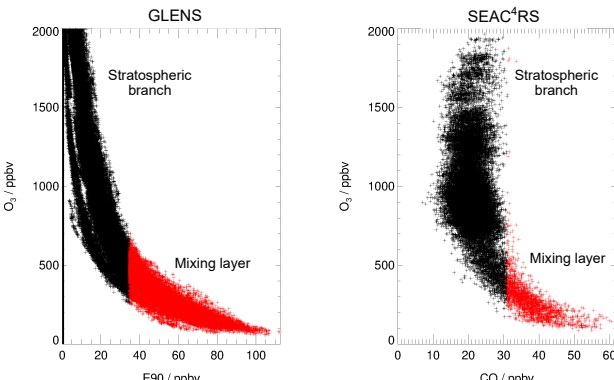

**Figure 2.** Tracer-tracer correlations for GLENS results in a latitude range from 30–35°N (left) and SEAC[4]RS measurements (right) consist of a stratospheric branch (black) and of the mixing layer between stratospheric and tropospheric air masses (red). The mixing layer is determined to be located above the tropopause and showing more than 35 ppbv E90 in GLENS and more than 31 ppbv CO in SEAC[4]RS measurements.

chlorine activation occurs is determined to be the temperature threshold for heterogeneous chlorine activation. The array of this temperature thresholds is dependent on a specific water vapour mixing ratio, which defined the chlorine activation threshold for the considered latitude, pressure and ozone range.

## 3 Analysing lowermost stratospheric GLENS results above central North America

The selected GLENS results are used as a data set representing the conditions and chemical composition in the mixing layer in the North American lowermost stratosphere in summer for all considered cases for future and today's conditions.

### 3.1 Comparing the GLENS mixing layer today with measurements

The reliability of the selected GLENS mixing layer of the C2010 case is analysed by comparing the GLENS mixing layer for the latitude range 30–35°N with the mixing layer derived from SEAC[4]RS ER2-aircraft measurements in August and September 2013. The SEAC[4]RS campaign was based in Houston (Texas) and one aim was to investigate the impact of deep convective clouds on the water vapour content in the lowermost stratosphere (Toon et al., 2016). Hence, SEAC[4]RS measurements represent moist and cold conditions enhancing the likelihood for heterogeneous chlorine activation to occur. Here, SEAC[4]RS trace gas measurements are used for CO (Harvard University Picarro Cavity Ring down Spectrometer (HUPCRS), Werner et al., 2017), $O_3$ (National Oceanic and Atmospheric Administration (NOAA) UAS $O_3$ instrument, Gao et al., 2012) and water vapour (Harvard Lyman-$\alpha$ photo fragment fluorescence hygrometer (HWV), Weinstock et al., 2009). Since GLENS is performed with a global model, the GLENS data cover a broader range in space (regarding altitude and area) than SEAC[4]RS





aircraft measurements, which were locally taken up to an altitude of 20 km. Hence, GLENS and SEAC$^4$RS air masses have a different spatial distribution in the lowermost stratosphere above North America.

The mixing layer between stratospheric and tropospheric air masses in the SEAC$^4$RS measurements is assumed to consist of measurements above the tropopause with a CO mixing ratio of more than 31 ppbv. This CO-boundary is selected to allow an

O$_3$-range similar to that of the GLENS mixing layer (up to ∼750 ppbv) and agrees with observations in the study by Pan et al. (2004), where mixed air masses between troposphere and stratosphere were described to hold usually more than ∼30 ppbv CO. In Fig. 2, the GLENS mixing layer (left) and the SEAC$^4$RS mixing layer (right) are marked in red while the stratospheric branch is shown in black. Air in the GLENS mixing layer is separated from tropospheric air by being located above the thermal tropopause and from stratospheric air by holding more than 35 ppbv of the E90 tracer. In the mixing layer deduced from

SEAC$^4$RS measurements, considered air masses lay above the tropopause as well and are separated here from the stratospheric branch by holding a CO mixing ratio of more then 31 ppbv.

Fig. 3 (top) shows the relative distribution of occurrence frequency of data points in the GLENS mixing layer (C2010) in the temperature-water vapour (left) and ozone-water vapour (right) correlation hereafter referred to as relative frequency distribution. For the relative frequency distribution in the temperature – water vapour correlation, the number of data points in the

GLENS mixing layer in all temperature and water vapour bins of the size of 1 K×1 ppmv H$_2$O (Fig. 3, left) are calculated considering the whole water vapour and temperature range given in Tab. 2. For the relative frequency distribution in the ozone-water vapour correlation (Fig. 3, right), the number of data points in all ozone and water vapour bins of the size 10 ppbv O$_3$× 1 ppmv H$_2$O are calculated. The number of data points of each temperature-H$_2$O (O$_3$-H$_2$O) bin is normalized by the total number of data points found in the GLENS mixing layer. These fractions are colour-coded in Fig. 3. The relative frequency

distribution of data points in the mixing layer derived from SEAC$^4$RS measurements in the same way is shown in Fig. 3 (bottom).

Comparing the SEAC$^4$RS and GLENS mixing layers yields a similar relative frequency distribution regarding temperature and H$_2$O conditions. Above 5 ppmv H$_2$O, the maximum fraction of GLENS and SEAC$^4$RS data resides in the same water vapour and temperature range of 201–207 K and 5–8 ppmv H$_2$O (Fig. 3, left). However, SEAC$^4$RS data show a higher fraction

at lower temperatures of 197–200 K and a higher fraction of GLENS data has lower water vapour mixing ratios than 5 ppmv. Furthermore, GLENS data spread over a broader water vapour range.

The SEAC$^4$RS and GLENS mixing layers show a similar distribution regarding the H$_2$O-O$_3$-correlation (Fig. 3, right). A significant fraction of all data corresponds to an ozone range of 200–350 ppbv, but in the GLENS data a higher fraction holds low water vapour mixing ratios with an ozone mixing ratio of 400–450 ppbv.

In addition to SEAC$^4$RS measurements, data in the GLENS mixing layer are compared with measurements sampled during the Stratosphere-Troposphere Analyses of Regional Transport (START08) campaign (Pan et al., 2010), which covers a larger latitude range over central North America than the SEAC$^4$RS measurements. The START08 campaign was designed to characterize the transport pathways in the extratropical tropopause region using the U.S. National Science Foundation (NSF) Gulfstream V (GV) research aircraft. START08 measurements show a good overall agreement with GLENS results, in spite

of the fact that a higher fraction of air masses sampled during START08 has temperatures higher than 215 K caused by the

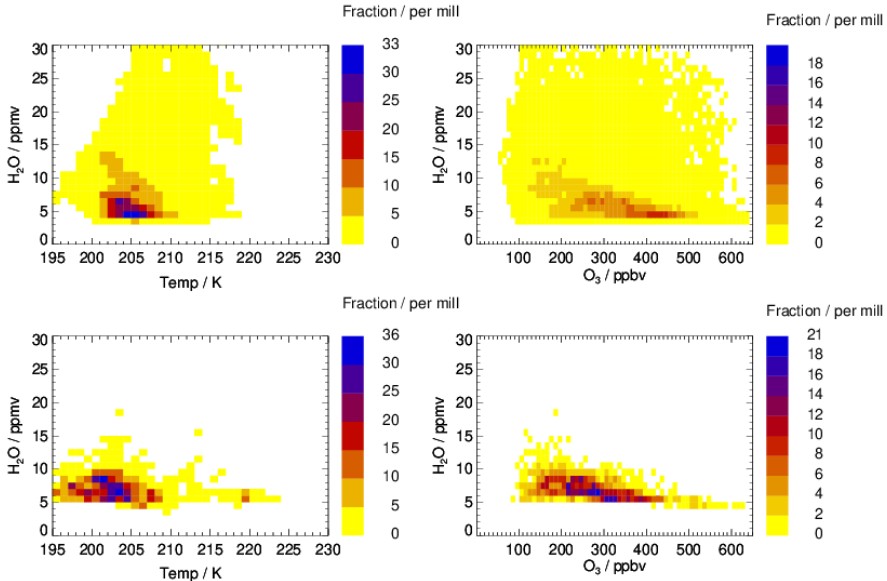

**Figure 3.** Comparison of the relative distribution of occurrence frequency of data points in the GLENS mixing layer (C2010) between stratospheric and tropospheric air masses (top) with measurements of the SEAC[4]RS aircraft campaign (bottom). Left panels show the relative frequency distribution regarding water vapour and temperature conditions and right panels regarding water vapour and ozone mixing ratios. The relative frequency distribution is derived by calculating the number of data points found in a specific temperature and water vapour bin (1 K × 1 ppmv $H_2O$, left) or ozone and water vapour bin (10 ppbv $O_3$ × 1 ppmv $H_2O$, right) considering all water vapour and temperature (ozone) ranges given in Tab. 2. The number of data points of each temperature-$H_2O$ ($O_3$-$H_2O$) bin is normalized by the total number of data points. The colour scheme marks these fractions.

maximum flight height of the GV of ~14.5 km (for more information see Appendix A).

In general, data points from GLENS representing the mixing layer have a good overall agreement with data points in the mixing layer deduced from aircraft measurements above North America. Measurements during SEAC[4]RS sampled convective injections penetrating water vapour into the stratosphere (Toon et al., 2016) and thus provide unusual cold and moist conditions for the lowermost stratosphere. Hence, the higher fraction of air masses describing temperatures of 197–200 K in the SEAC[4]RS measurements is considered as a case study in Sec. 4.5.

### 3.2 Change in the chemical composition of the mixing layer

The chemical composition of the mixing layer will change in the GLENS future scenarios. In Fig. 4, the E90-$O_3$-correlation is shown for all considered cases (see Tab. 1). In the climate change scenario (C2040, C2090), the $O_3$ mixing ratio increases during the 21st century, but the ozone mixing ratio in the geoengineering scenario (F2040, F2090) remains in a similar range of ~200–600 ppbv as in case C2010. The correlation between ozone and the artificial tropospheric tracer E90 for C2010 (grey),



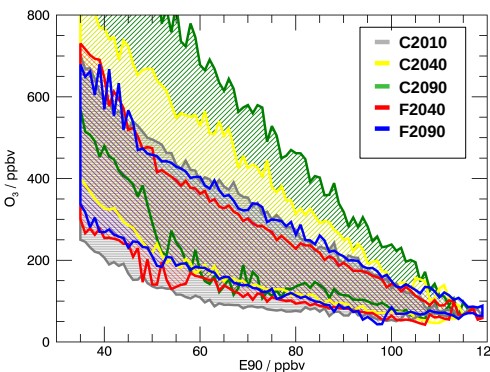

**Figure 4.** $O_3$-E90-correlation in the GLENS mixing layer for today (C2010) and the future scenarios considering both a climate change (C2040, C2090) and additional geoengineering (F2040, F2090). An overview on the presented cases is given in Tab. 1.

shown in Fig. 4, agrees well with the F2040 case (red) and the F2090 case (blue). For cases with climate change, the ozone mixing ratio is significantly higher in case C2040 (yellow) and C2090 (green) especially for low E90 concentrations. The enhancement of ozone in the mixing layer could be related to a higher ozone smog production because of a higher atmospheric $CH_4$ burden with increasing $CH_4$ emissions in the RCP8.5 scenario. Furthermore, climate change is expected to increase upper

stratospheric ozone and accelerate the BDC transporting more ozone from high altitudes downwards in the lowermost stratosphere (Iglesias-Suarez et al., 2016).

Besides changes in transport, ozone in the mid-latitude mixing layer could be affected by changes in chemistry (e.g. through chlorine activation). The conditions causing heterogeneous chlorine activation are determined first of all by temperature and water vapour mixing ratios. Furthermore, $Cl_y$- and $NO_y$-mixing ratios affect the threshold between conditions, which may or

may not lead to chlorine activation. The distribution of temperatures and several trace gas mixing ratios within the GLENS mixing layer is shown in Fig. 5 for the subtropical (30–35°N) and the extra-tropical (44–49°N) latitude band over central North America.

In all future scenarios, temperatures and water vapour mixing ratios increase (Fig. 5). In the subtropical latitude band, the median temperature increases from today to the end of the 21st century by ~3 K assuming a climate change scenario and by

~5.5 K when applying geoengineering. In the extra-tropical latitude band, the temperature is higher and shows a similar increasing trend. Water vapour mixing ratios are higher in the extra-tropical latitude band than in the subtropical band and spread over a broader range. In both latitude ranges and future scenarios the water vapour content increases until the end of the 21st century driven by increasing temperatures of the tropical tropopause layer. An increase in water vapour enhances $HO_x$-mixing ratios (Fig. 5) and thus accelerates ozone destruction in the $HO_x$-cycle.

The HCl and $ClO_x$ mixing ratios decrease in the GLENS simulations for future scenarios due to the implementation of boundary conditions in the WACCM according to the Montreal Protocol and its amendments and adjustments. However, the median





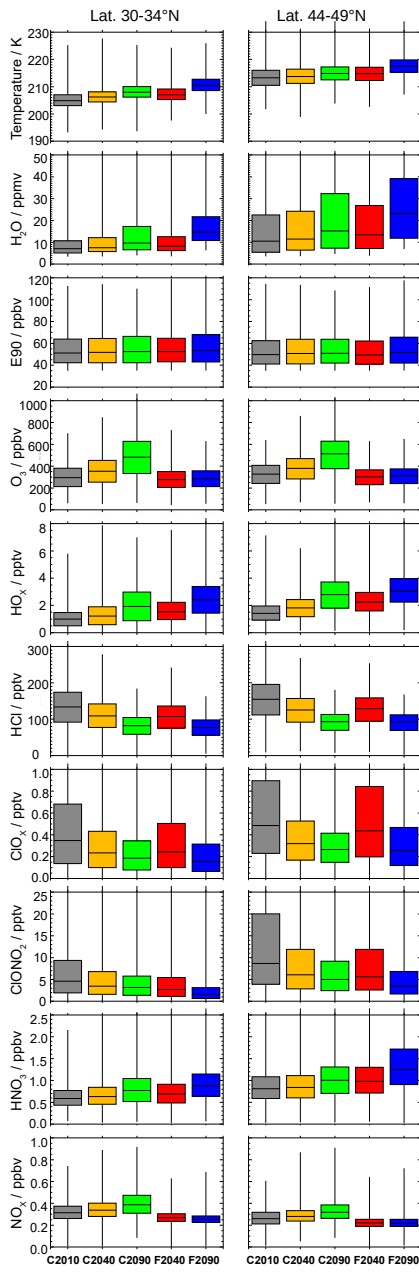

**Figure 5.** Distribution of temperatures and several trace gas mixing ratios in the GLENS mixing layer for case C2010 and future scenarios considering both climate change (C2040, C2090) and in addition sulfate geoengineering (F2040, F2090) (see Tab. 1). The frequency distribution is illustrated as box-plots, where the upper and lower quartile (75% and 25%) of the data set is marked by the upper and lower end of the box. The median of the temperature or mixing ratio values within the mixing layer is illustrated through the horizontal line in the box. Ends of vertical lines mark the minimum and the maximum value of the considered data.





$ClO_x$ mixing ratio is higher by $\sim 8$ (30–34°N) – 22% (44–49°N) in the F2040 case than in the C2040 case. This could be due to a reduced $NO_x$ mixing ratio in the F2040 case. In both future scenarios, the $HNO_3$ mixing ratio increases until the year 2100 (Fig. 5). For climate change, the $NO_x$ mixing ratio increases as well. It decreases in the geoengineering scenario, because $HNO_3$ formation is accelerated through heterogeneous reactions favoured by a higher aerosol abundance and increasing

temperatures enhancing the $NO_2/NO$-ratio. Less $NO_x$ causes less $ClO_x$ to be bound in $ClONO_2$, thus resulting in more gas phase $ClO_x$ in the geoengineering scenario. Additionally, the occurrence of heterogeneous chlorine activation could yield an enhancement of $ClO_x$ in the geoengineering scenario due to an enhanced aerosol abundance.

The changes in chemistry may affect the future ozone abundance in the lowermost stratosphere. The median ozone mixing ratio increases by 60–67% until the year 2100 in the climate change scenario but remains at today's level in the geoengineering

scenario (Fig. 5). The partitioning between active radicals ($ClO_x$, $NO_x$) and reservoir species (HCl, $HNO_3$) differs between the climate change (C2040, C2090) and the geoengineering (F2040, F2090) cases resulting in a different chemical impact on ozone. The likelihood for the occurrence of ozone loss caused by heterogeneous chlorine activation may differ as well in the future scenarios, because the heterogeneous chlorine activation is stronger for low temperatures and enhanced water vapour mixing values. The likelihood of heterogeneous chlorine activation to occur and its impact on the ozone chemistry is analysed

below in the subsequent section.

## 4    Comparison of GLENS results with chlorine activation thresholds

The water vapour and temperature range, in which heterogeneous chlorine activation occurs, is determined by calculating chlorine activation thresholds for the specific chemical composition using the CLaMS model. The fraction of all air masses in the GLENS mixing layer between the troposphere and the stratosphere with conditions leading to chlorine activation accounts

for the likelihood that chlorine activation occurs in the considered cases. The chlorine activation threshold is determined based on the composition of air masses in the mixing layer between tropospheric and stratospheric air deduced from GLENS results (see Sec. 3). Chlorine activation thresholds are calculated for all cases (see Tab. 1) with CLaMS (see Sec. 2.2) for 4 latitude ranges from 30–49°N, 5 pressure ranges between 70 and 300 hPa and 5 different ozone ranges from 150–650 ppbv (see Tab.2). Ozone values lower than 150 ppbv are not considered here, because only a minor fraction of air parcels shows less than 150

ppbv ozone. Furthermore, a critical ozone amount has to be exceeded for chlorine activation to occur (von Hobe et al., 2011), because a higher ozone mixing ratio causes a higher ClO/Cl-ratio and thus more $ClONO_2$ formed. This is important for heterogeneous chlorine activation in reaction R1 to occur.

### 4.1    Analysis of chlorine activation thresholds

Both the chlorine activation threshold and the $H_2O$-temperature relative frequency distribution vary depending on the assumed

pressure and ozone level and thus for different data groups. An example for the impact of the pressure and ozone range on the $H_2O$-temperature relative frequency distribution and the chlorine activation threshold is shown in Fig. 6 for the GLENS mixing layer in the latitude range of 30–35°N of the C2010 case.

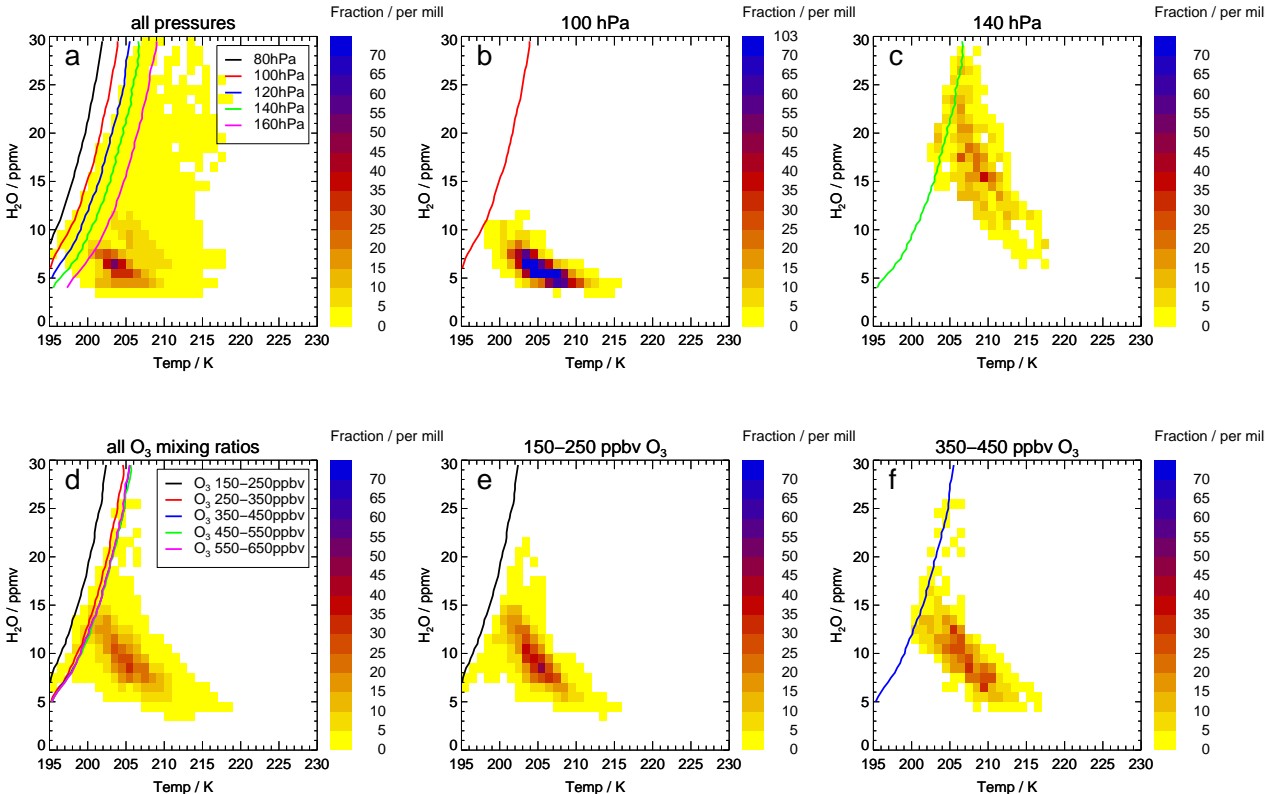

**Figure 6.** $H_2O$-temperature relative frequency distributions and chlorine activation thresholds of different data groups (see Tab. 2) for the GLENS C2010 case and a latitude range of 30–35°N. The $H_2O$-temperature relative frequency distribution is illustrated as a colour scheme. The colour marks the fraction of the considered data corresponding to a water vapour and temperature bin (1 ppmv $H_2O \times 1$ K). The water vapour and temperature dependent chlorine activation thresholds are marked as a line. Top panels are related to data groups with an ozone mixing ratios of 350–450 ppbv: all data in the considered latitude and ozone range (30–35°N, 350–450 ppbv $O_3$) (a); the data group defined by a latitude of 30–35°N, an ozone mixing ratio of 350–450 ppbv $O_3$ and the 100 hPa pressure level (b); and the data group defined by a pressure level of 140 hPa and the same latitude and ozone range (c). Bottom panels are related to data groups with a pressure level of 120 hPa: all data in the considered latitude and pressure level (30–35°N, 120 hPa) (d); the data group defined by a latitude of 30–35°N, a pressure level of 120 hPa and an ozone mixing ratio of 150–250 ppbv $O_3$ (e); and the data group defined by an ozone mixing ratio of 350–450 ppbv and the same latitude and pressure level (f).

The $H_2O$-temperature relative frequency distribution is shown (Fig. 6, top) for an ozone range of 350–450 ppbv. The water vapour and temperature dependent chlorine activation thresholds are marked as a line for different pressure levels (see Tab. 2). In Fig. 6 a, chlorine activation thresholds are plotted for all pressure levels in the considered latitude and ozone range (30–35°N,





350–450 ppbv $O_3$).

At higher pressure levels (lower altitudes), the chlorine activation threshold is shifted allowing chlorine activation to occur at higher temperatures (Fig. 6 a). This shift is due to an increasing liquid particle formation as well as more $ClONO_2$ absorbed by an aerosol particle at higher pressures. The heterogeneous chlorine activation rate of reaction R1 is determined by the $ClONO_2$

uptake into the aerosol particle (Shi et al., 2001). Air masses lying on the left side of the chlorine activation threshold show chlorine activation. The relative frequency distribution shown in Fig. 6 a, is related to all air masses with 350–450 ppbv ozone in a latitude range of 30–35°N. Some data points cross various activation thresholds. However, only data points crossing the chlorine activation threshold and in addition corresponding to the pressure level of the activation threshold will yield activated chlorine. As an example, the chlorine activation thresholds at the 100 hPa and 140 hPa level are plotted together with

the GLENS relative frequency distribution corresponding to the same data group (Fig. 6 b, c). Air masses in the 100 hPa level (Fig. 6 b) are colder and dryer than those at 140 hPa (Fig. 6 c). Hence, at the 100 hPa level no chlorine will be activated (there are no data corresponding to a $H_2O$-temperature bin on the left side of the threshold line) and chlorine activation occurs for the 140 hPa level only for data points with a high water vapour mixing ratio.

In Fig. 6 (bottom), the $H_2O$-temperature relative frequency distribution and the chlorine activation thresholds are presented for

a pressure level of 120 hPa. The impact of the ozone mixing ratios on the chlorine activation threshold is illustrated. Fig 6 d shows the GLENS $H_2O$-temperature relative frequency distribution and the chlorine activation thresholds for all data groups corresponding to the selected latitude range and pressure level (30–35°N, 120 hPa). Higher ozone mixing ratios are related to higher $Cl_y$ amounts. Hence, an increase in ozone shifts the chlorine activation threshold to higher temperatures (Fig. 6 d). However, considering the relative frequency distribution of specific data groups with different ozone levels, data points with

more ozone are warmer than those with less ozone (Fig. 6 e, f).

In the future scenarios, the $H_2O$-temperature relative frequency distribution as well as the chlorine activation thresholds vary. In Fig. 7 the $H_2O$-temperature relative frequency distribution is shown for the cases C2010, C2090 and F2090. The relative frequency distributions are shown for the subtropical latitude band (30–35°N, Fig. 7 a–c) and for extra-tropical latitude band from (44–49°N, Fig. 7 d–f). For each case shown, additionally a selection of chlorine activation thresholds is shown. These

are related to different ozone and pressure levels and give a range of uncertainty for the water vapour and temperature ranges causing chlorine activation.

In agreement with the changes of the conditions in the mixing layer described in Sec. 3, the future $H_2O$-temperature relative frequency distributions (C2090 in Fig. 7 b, F2090 in Fig. 7 c) are both moister and warmer than the conditions today (C2010, Fig. 7 a). However, the geoengineering case F2090 exhibits data significantly warmer and moister than reached in the climate

change case C2090. In the extra-tropical latitude band (Fig. 7 d–f), GLENS results are generally warmer than in the subtropical latitude range.

Considering the chlorine activation thresholds in Fig. 7, the largest fraction of air masses corresponds to temperatures greater than the chlorine activation thresholds. The chlorine activation thresholds for the C2090 case are shifted to lower temperatures compared to case C2010, because of the lower chlorine abundance (a higher $Cl_y$ mixing ratio promotes heterogeneous chlorine

activation (Robrecht et al., 2019)). In contrast, in the geoengineering scenario F2090 chlorine activation can occur at higher

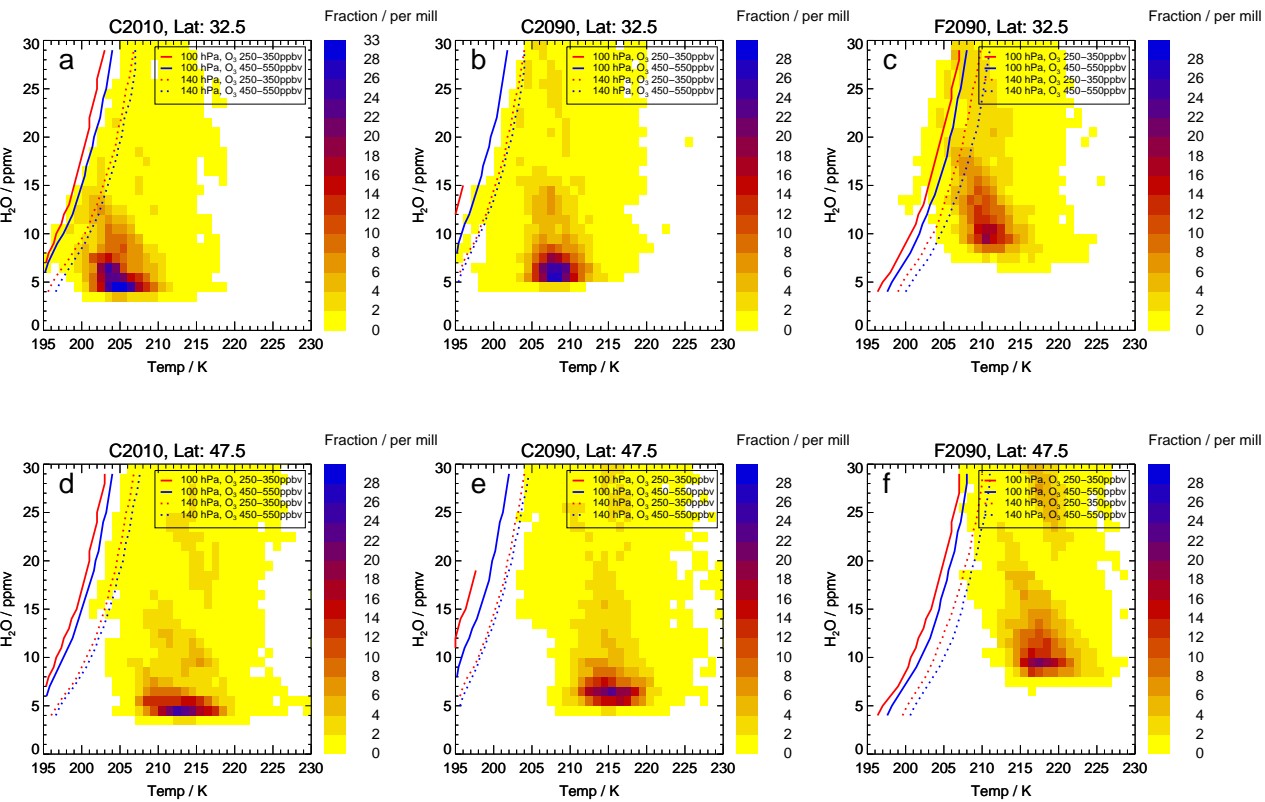

**Figure 7.** $H_2O$-temperature relative frequency distributions and examples for chlorine activation thresholds for the cases C2010 (a, d) and the future scenarios at the end of the 21st century assuming climate change (C2090, b, e) and additional geoengineering (F2090, c, f) for the subtropical latitude band (30–35°N, top) and the extra-tropical (44–49°N, bottom). The colour marks the fraction of the considered data corresponding to a water vapour and temperature bin (1 ppmv $H_2O \times 1$ K). The water vapour and temperature dependent chlorine activation threshold is marked as a line for exemplarily chosen data groups specified in the legend of each panel.

temperatures than today in spite of the lower chlorine amount. This is caused by the higher aerosol loading due to the applied geoengineering.

In each case, the water vapour and temperature bins marked by the chlorine activation thresholds to potentially cause heterogeneous chlorine activation are in good agreement for both latitude ranges presented. Since the temperatures of the mixing
5 layer are higher in the extra-tropical latitude band, the fraction of air masses crossing the chlorine activation threshold and thus causing chlorine activation is lower in that latitude range (44–49°N) than in the subtropical latitude band (30–35°N).

There are some chlorine activation thresholds that cannot be reported when the water vapour mixing ratio exceeds a cer-





tain value (e.g. Fig. 7 e, 100 hPa, 250–350 ppbv $O_3$). At such high water vapour mixing ratios, HCl is absorbed strongly into the aerosol particles, reducing gas phase $Cl_y$ and thus less $ClONO_2$ may be formed. Since chlorine is activated in R1 (HCl+$ClONO_2$), less $ClONO_2$ leads to a lower chlorine activation rate. This effect is negligible if the $Cl_y$ mixing ratio is high enough. But if the $Cl_y$ mixing ratio is low (e.g. in a low ozone range in the years 2090–2099), reducing gas phase $Cl_y$ by

absorbing HCl into the aerosol results in no activation of chlorine. Hence, there is no chlorine activation for these conditions. Summarizing, the water vapour and temperature dependent chlorine activation threshold marks an upper boundary of temperatures causing heterogeneous chlorine activation for air masses with a specific water vapour mixing ratio. Thus for a given water vapour mixing ratio, the maximum temperature at which chlorine activation may occur is determined by the chlorine activation threshold. In this section, we showed that the chlorine activation thresholds and the $H_2O$-temperature relative frequency

distribution of the GLENS mixing layer depend on the aerosol abundance, pressure and the $Cl_y$ mixing ratio, which is related to the ozone level. Moist and very cold air masses, which in general are expected to promote heterogeneous chlorine activation, usually correspond to low pressures and low ozone mixing ratios. Hence, the pressure and ozone dependence of chlorine activation results in only few air masses with conditions suitable to activate chlorine. Thus, chlorine activation thresholds have to be compared with air masses in GLENS corresponding to the same data group regarding pressure, ozone and latitude range

as the calculated chlorine activation threshold to deduce the likelihood that chlorine activation occurs.

### 4.2 Likelihood for ozone destruction today and in future

The likelihood for chlorine activation to occur is quantified here as the fraction of air masses in the GLENS mixing layer between tropospheric and stratospheric air, which are cold and moist enough to cause heterogeneous chlorine activation. Comparing GLENS air masses with chlorine activation thresholds, the number of air masses is counted showing lower temperatures

than determined as the threshold temperature for chlorine activation. The fraction of this amount in all air masses within the GLENS mixing layer yields the likelihood for heterogeneous chlorine activation to occur. Here, we assume that chlorine activation always results in ozone destruction processes known from polar late winter and early spring (e.g. Molina and Molina, 1987; McElroy et al., 1986; Crutzen et al., 1992; Solomon, 1999). Hence, the likelihood for chlorine activation to occur is the same as the likelihood for chlorine catalysed ozone destruction.

In Fig. 8 (top), the likelihood for chlorine activation to occur is presented considering air masses in the entire latitude range (30–49°N) of the GLENS mixing layer. Each panel corresponds to a considered case (C2010, C2040, C2090, F2040 and F2090, see Tab. 1). The likelihood for chlorine activation to occur is marked by the height of a bar: for single pressure levels and named 'all' for all air masses within the mixing layer. In the C2010 case, the overall likelihood for chlorine activation to occur is 1.0% in the entire latitude and pressure level (Fig. 8, top, left panel, left bar). However, chlorine activation occurs most

likely in the pressure level of 140 hPa. A fraction of 3.5% of all air masses in the 140 hPa level causes heterogeneous chlorine activation in the C2010 case. As described in Sec. 4.1, higher pressures increase the aerosol formation and uptake of $ClONO_2$ into the liquid aerosol particles, which determines if chlorine activation through reaction R1 (Shi et al., 2001) occurs. Thus, the chlorine activation threshold is shifted to higher temperatures at higher pressures. However, the likelihood for chlorine activation deduced from the comparison between chlorine activation thresholds and air masses in the GLENS mixing layer is lower





at 160 hPa than at 140 hPa (Fig. 8), because air masses corresponding to higher pressure levels are warmer than those with a lower pressure (exemplary shown in Fig. 6 b, c). Air masses in the 160 hPa level are significantly warmer than air masses in the 140 hPa level. Hence, although the chlorine activation threshold is shifted to higher temperatures for the 160 hPa pressure level, most air masses corresponding to this high pressure are to warm for heterogeneous chlorine activation and chlorine activation

occurs most likely in the 140 hPa level.

The contribution of different ozone levels in the air masses, which show chlorine activation, is additionally marked by the colour scheme in Fig. 8. In case C2010, chlorine activation mainly occurs in air masses with an ozone mixing ratio of 250–350 ppbv (Fig. 8 top).

Focussing on the future scenarios, the likelihood for chlorine activation to occur is very low in the climate change cases C2040

and C2090 (Fig. 8 top). In contrast, the likelihood in the geoengineering cases F2040 and F2090 is higher than for today (case C2010). Chlorine activation occurs most likely at the mid of the 21st century in case F2040, where 3.3% of all air masses in the GLENS mixing layer would cause chlorine activation. 11.5% of the air masses in the 140 hPa pressure level cause chlorine activation in case F2040. The likelihood for chlorine activation to occur is slightly lower at the end of the 21st century due to the decrease of $Cl_y$ implemented in GLENS. In case F2090, 2.7% of all air masses in the GLENS mixing layer cause chlorine

activation.

The likelihood for chlorine activation in different latitude ranges is illustrated in Fig. 8 in the middle (latitude range of 30–35°N) and bottom panels (latitude range of 44–49°N). In general, chlorine activation occurs more likely in the subtropical latitude band (30–35°N) than in extra-tropical (44–49°N) latitudes, because of the different temperature range and chemical composition around the tropopause in the tropics and extra-tropics (note the different y-scales for different latitude ranges in

Fig. 8). In case C2010, 1.1% of all GLENS air masses in the subtropical latitude band (30–35°N) causes chlorine activation and 0.9% in the extra-tropical latitude band (44–49°N). In both latitude ranges, the likelihood for chlorine activation is negligible in the future cases C2040 and C2090. In contrast, the likelihood increases in the geoengineering scenario. In case F2040, 4.1% of all air masses in the subtropical latitude band of the considered GLENS mixing layer cause chlorine activation. In the same latitude range, the likelihood for chlorine activation to occur is higher in case F2090 (4.5%), in spite of the implemented de-

crease in stratospheric $Cl_y$. The likelihood increases between case F2040 and F2090, because in case F2090 a higher fraction of air masses has a pressure corresponding to the 120 hPa and the 140 hPa level than in case F2040 (not shown). In contrast, in the extra-tropical latitudes (44–49°N), the likelihood for chlorine activation to occur is higher in case F2040 (1.3%) than in case F2090 (0.2%) caused by the decrease in stratospheric $Cl_y$ and the warming of the mixing layer. In this latitude range, the likelihood for chlorine activation to occur is generally lower than in the subtropical latitude band, because the temperatures in

the GLENS mixing layer are higher (see Fig. 5).

Focussing on the ozone mixing ratio of air masses in which chlorine activation occurs in the GLENS mixing layer, the colour scheme in Fig. 8 indicates that chlorine activation occurs more likely in air masses with low ozone mixing ratios than in air masses with high ozone mixing ratios. This is in agreement with the dependence of the $H_2O$-temperature relative frequency distribution in the GLENS mixing layer on the ozone mixing ratio discussed in Sec. 4.1 (exemplary shown in Fig. 6). Air

masses with higher ozone mixing ratios are warmer than those with less ozone and thus cause less likely heterogeneous chlo-





**Figure 8.** Likelihood for heterogeneous chlorine activation to occur in different latitude regions in the GLENS mixing layer for all considered cases of today and the future scenarios (see Tab. 1). The entire latitude range above central North America (30–49°N) is considered (top), only the subtropical latitude band (30–35°N) (middle) and only the extra-tropical latitude band (44–49°N) (bottom). Different panels correspond to different cases given at the top of each panel. The height of the bars marks the likelihood for a specific pressure level given under that bar. The pressure range corresponding to a given pressure level is given in Tab. 2. The denotation *all* refers to the whole pressure range of the mixing layer. Colours indicate the likelihood for chlorine activation to occur for air masses with different ozone ranges. Note the changes scale of the y-axis in different rows.

rine activation.

In summary, the occurrence of chlorine activation and the resulting catalytic ozone loss processes similar to those known from polar regions is unlikely based on the comparison of GLENS results with chlorine activation thresholds. However, chlorine activation occurs more likely in the future scenario assuming geoengineering than in today's case C2010. In the future scenario assuming climate change, the likelihood for chlorine activation to occur is negligible. Furthermore, chlorine activation is more





likely in lower latitudes than in higher latitudes. Since air masses causing chlorine activation usually show low ozone mixing ratios, the ozone amount affected by chlorine catalysed ozone destruction is expected to be low. How relevant the activation of chlorine is for the ozone chemistry in the mid-latitude lowermost stratosphere is analysed in the next section.

### 4.3    Impact of heterogeneous chlorine activation on ozone in the lowermost stratosphere

How much ozone in the mixing layer above central North America is affected by the heterogeneous chlorine activation process is analysed here by considering the ozone changes in the CLaMS simulations together with the relative frequency distribution in the temperature-water vapour correlation in GLENS. Briefly, ozone changes in CLaMS correspond to an upper boundary for the conditions assumed during the simulation and the relative frequency distribution comprises the fraction of data points with the same water vapour and temperature conditions as assumed during the simulation. Hence, from combining the ozone

change in CLaMS simulations with the relative frequency distribution in GLENS, the impact of this ozone loss process on ozone in the mixing layer can be determined.

In more detail, CLaMS simulations are conducted for all data groups and any combination of temperature and water vapour bins. The difference between initial and final ozone within each 10 day simulation yields for each data group (determined by a latitude, pressure and ozone range, see Tab. 2) the chemical ozone change corresponding to a particular water vapour

and temperature bin ($1\,\mathrm{K} \times 1\,\mathrm{ppmv}\;H_2O$ in a range of 195–230 K and 4–30 ppmv $H_2O$). Since no mixing is allowed in the box-model runs, in this way conditions, which yield chlorine activation, are not disturbed within 10 days. In the lowermost stratosphere, the duration of maintenance for conditions causing chlorine activation is not yet known. However, mixing of cold and moist air from the troposphere uplifted to above the tropopause (e.g. through convective overshooting) with dry and warmer stratospheric air will reduce the water vapour content of the moist air parcel. Since the occurrence of chlorine

activation depends on both the temperature and the water vapour mixing ratio of the air parcel, a decrease in water vapour can stop chemical chlorine activation. Hence, assuming the maintenance of chlorine activation for 10 days without a perturbation by mixing here, yields an upper boundary for the impact of heterogeneous chlorine activation on ozone in the mid-latitude lowermost stratosphere.

The chemical ozone change between the initial and final ozone of a 10 day box-model simulation is multiplied with the

number of GLENS air masses corresponding to the same data group and water vapour and temperature bin. In this way, the total chemical ozone change in the mid-latitude mixing layer is estimated. The total initial ozone is calculated by multiplying the median ozone amount of each data group with the number of GLENS air masses corresponding to that data group. The ratio of the total ozone change from the start to the end of the 10 day CLaMS simulation and the total initial ozone yields the relative ozone change.

The relative ozone change in the GLENS mixing layer determined from the difference between final and initial ozone in the 10 day CLaMS box-model simulatios for each considered case is illustrated as black bars in Fig. 9 (top). In case C2010, chemical ozone formation dominates the ozone chemistry in the mixing layer and causes an increase in ozone of 2.3%. In the future climate change scenario, ozone would increase by around 2.5% within 10 days of unperturbed chemistry in case C2040 and by 3% in case C2090. This increasing ozone in the future may be related to the reduction of ODS implemented in GLENS. In the





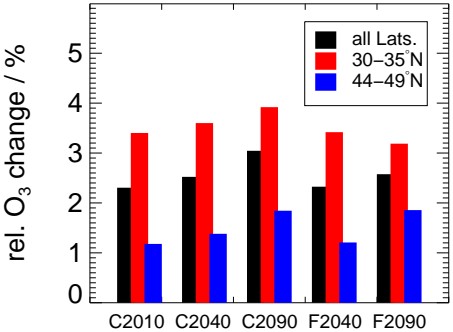

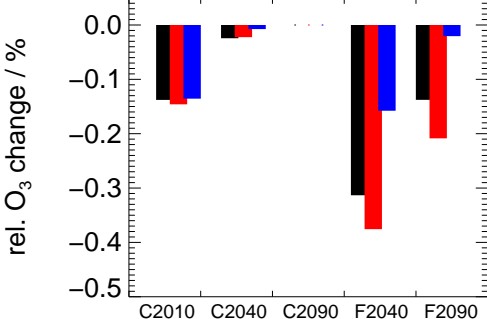

**Figure 9.** Relative chemical ozone change in the mixing layer determined from the difference between initial and final ozone in 10 day CLaMS box-model simulations (no mixing between air masses) in all considered cases (see Tab. 1). The relative ozone change is shown (top) considering the entire latitude region above central North America (black bars) as well as only considering the subtropical (30–35°N, red bars) or extra-tropical (44-49°N, blue bars) latitude region. Further, the ozone change from air masses in which chlorine activation can occur normalized by the total initial ozone from all air masses in the GLENS mixing layer is shown (bottom).

geoengineering scenario, the relative chemical ozone formation is lower than following a climate change. However, the ozone change increases from +2.3% in the F2040 case to +2.6% in the F2090 case. The lower chemical ozone increase in the mixing layer for the geoengineering scenario is based on an increase of ozone destruction processes. Ozone destruction catalysed by $HO_x$-radicals is more likely in the geoengineering scenario because of the higher $HO_x$ mixing ratio (Fig. 5). Furthermore

5   heterogeneous chlorine activation could yield ozone destruction.

The relative ozone change caused by heterogeneous chlorine activation is shown in Fig. 9 (bottom). For calculating the relative ozone change caused by heterogeneous chlorine activation, ozone changes corresponding to air masses which cause chlorine activation are multiplied with the number of these air masses in the GLENS mixing layer. This ozone change from air masses



in which chlorine activation can occur is normalized with the total initial ozone of all air masses in the GLENS mixing layer. Black bars correspond to air masses in the entire latitude region above central North America. In case C2010, 0.1% of ozone in the mixing layer would be destroyed within 10 days caused by heterogeneous chlorine activation. In the climate change scenarios, chlorine activation causes less ozone destruction in the mixing layer. Heterogeneous chlorine activation has the

strongest impact on ozone in the GLENS mixing layer of the F2040 case. In this case, 0.3% of ozone in the mixing layer would be destroyed, if the chemical conditions yielding chlorine activation are maintained for 10 days. In comparison, for the conditions in case F2090 0.1% of ozone in the mixing layer would be destroyed.

In Fig. 9, additionally the relative ozone change calculated based on 10 day CLaMS box-model simulations is illustrated with respect to the latitude ranges 30–35°N (red bars) and 44–49°N (blue bars). Comparing the relative ozone change in different

latitude regions, in the subtropical latitude band more ozone is formed (Fig. 9 top). For example in case C2010, ozone increases by 3.4% in the subtropical latitude band (30–35°N) and by 1.2% at 44–49°N. However, in the subtropical latitude band heterogeneous chlorine activation affects ozone more (Fig. 9, bottom). Heterogeneous chlorine activation causes the strongest ozone destruction in the mixing layer for the geoengineering case F2040 with an ozone destruction of 0.4% in 10 days in 30–35°N. In contrast, in case C2040 less than 0.1% would be destroyed in the same latitude range.

Since in the subtropical latitude range (30–35°N) the effect of heterogeneous chlorine activation on ozone in the highest, the relative ozone change in that latitude range is shown more detailed with respect to single pressure levels in Fig. 10. In general, ozone formation processes dominate at low pressures, causing a net chemical ozone increase (Fig. 10, top). In higher pressure levels, the net ozone formation is lower. Furthermore, the occurrence of heterogeneous chlorine activation is more likely at higher pressure levels. In the F2040 case where heterogeneous chlorine activation has the strongest impact on ozone chemistry

in the mixing layer, up to 2.1% of total initial ozone in the pressure level of 140 hPa are destroyed in air masses with conditions allowing heterogeneous chlorine activation (Fig. 10, bottom). This results in a net ozone change in this pressure level of $-0.7\%$ (Fig. 10, top), when both the ozone destruction in air masses allowing chlorine activation and ozone formation in the other air masses in the 140 hPa level are considered.

The likelihood for chlorine activation to occur in the mid-latitude mixing layer just above the tropopause, the relative ozone

change caused by heterogeneous chlorine activation and the net chemical ozone change in the GLENS mixing layer determined from 10 day CLaMS box-model simulations is summarized in Tab. 3 considering the entire latitude range (30–49°N) as well as for the subtropical (30–35°N) and the extra-tropical (44–49°N) latitude band. The results calculated here are referred to as 'reference'.

In general, the impact of heterogeneous chlorine activation causing chlorine catalysed ozone destruction on ozone in the mid-

latitude lowermost stratosphere is low. Combining the occurrence of conditions in GLENS with the chemical ozone change determined through CLaMS box-model simulations, in all cases a net chemical ozone formation will occur above central North America. However, chlorine activation may affect ozone in the mixing layer. In the geoengineering scenario in case F2040 chlorine activation has the highest impact on ozone in comparison to the other cases and can cause an ozone reduction of up to 0.38%. However, the ozone changes determined in this study give an upper limit for ozone change caused by heterogeneous

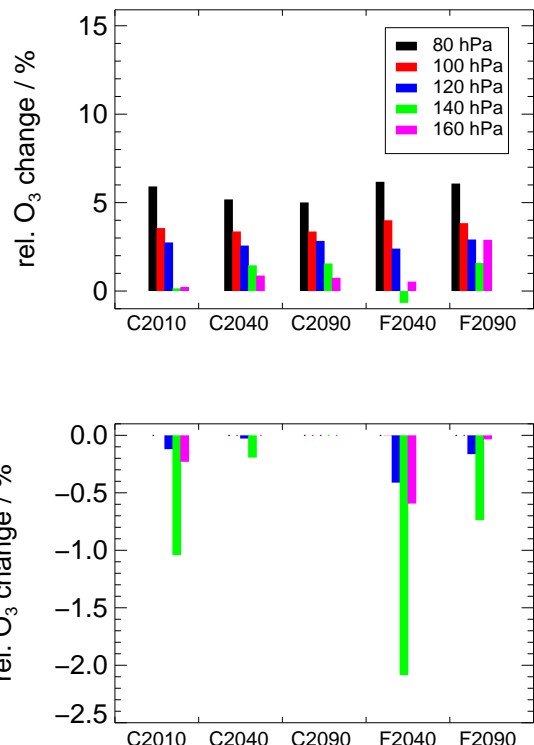

**Figure 10.** Relative chemical ozone change in the subtropical latitude band (30–35°N) in the mixing layer determined from the difference between initial and final ozone in 10 day CLaMS box-model simulations in the considered cases (see Tab. 1). The overall ozone change (top) within single pressure levels between 70 and 300 hPa (see Tab. 2) is shown as well as the ozone change from air masses in which chlorine activation can occur normalized by the total initial ozone from all air masses in the GLENS mixing layer (bottom).

chlorine activation, because mixing between air masses is neglected in the box-model simulations used to calculate the chemical ozone change.

## 4.4 Relevance of heterogeneous chlorine activation in the mixing layer for the mid-latitude ozone column

In the previous section, an upper limit for ozone reduction caused by heterogeneous chlorine activation in the mid-latitude
5 mixing layer between tropospheric and stratospheric air was determined. Based on this relative ozone change in the mixing layer, the impact of heterogeneous chlorine activation in the mid-latitude lowermost stratosphere on column ozone is deduced. For all of the GLENS cases today and in future (see Tab. 1), first the ozone profile is determined by averaging over the ozone mixing ratio within each GLENS vertical level. Both the entire latitude region above central North America and specific lati-





**Table 3.** Overview on the likelihood for chlorine activation to occur in the mid-latitude mixing layer above the tropopause, its impact on ozone in the mixing layer and the relevance for ozone column. Further the net chemical ozone change in the mixing layer is specified. Three latitude ranges are considered here: 30–49°N, only the subtropical latitude band in 30–35°N and only the extra-tropical latitude band in 44–49°N. The considered cases today (C2010) and in the future scenarios assuming a climate change (C2040, C2090) and additional geoengineering (F2040, F2090) are further described in Tab. 1. The reference refers to results deduced from the GLENS mixing layer. In the assumption with 2 K lower temperatures, temperatures of GLENS air masses are reduced of 2 K to infer uncertainties in GLENS temperatures. The chemical ozone changes here mark an upper limit for the impact of heterogeneous chlorine activation in the mixing layer on ozone, because ozone changes are determined based on 10 day box-model simulations neglecting mixing between neighbouring air masses. Thus conditions causing chlorine activation are assumed here to be maintained for 10 days without perturbations.

| | $O_3$-Column / DU | $O_3$-Column in the mixing layer / DU | Reference | | | | 2 K lower temperatures | | | |
| --- | --- | --- | --- | --- | --- | --- | --- | --- | --- | --- |
| | | | Likelihood for chlorine activation | net ozone change | rel. $O_3$-loss | $O_3$-loss / DU | Likelihood for chlorine activation | net ozone change | rel. $O_3$-loss | $O_3$-loss / DU |
| **C2010** | | | | | | | | | | |
| all Lats. | 295.8 | 15.7 | 1.0% | 2.3% | 0.1% | 0.02 | 3.7% | 1.8% | 0.5% | 0.08 |
| 30–35°N | 289.9 | 14.2 | 1.1% | 3.4% | 0.2% | 0.02 | 4.4% | 2.8% | 0.5% | 0.07 |
| 44–49°N | 307.8 | 20.3 | 0.9% | 1.2% | 0.1% | 0.03 | 2.4% | 0.9% | 0.4% | 0.08 |
| **C2040** | | | | | | | | | | |
| all Lats. | 307.2 | 16.9 | 0.1% | 2.5% | <0.1% | <0.01 | 1.4% | 2.4% | 0.1% | 0.02 |
| 30–35°N | 299.0 | 13.9 | 0.1% | 3.6% | <0.1% | <0.01 | 2.1% | 3.3% | 0.2% | 0.03 |
| 44–49°N | 319.2 | 21.4 | 0.1% | 1.4% | <0.1% | <0.01 | 0.5% | 1.3% | 0.1% | 0.01 |
| **C2090** | | | | | | | | | | |
| all Lats. | 321.7 | 18.8 | 0.0% | 3.0% | 0.0% | 0.00 | 0.2% | 3.0% | <0.1% | <0.01 |
| 30–35°N | 309.6 | 15.5 | 0.0% | 3.9% | 0.0% | 0.00 | 0.3% | 3.9% | <0.1% | <0.01 |
| 44–49°N | 336.9 | 23.8 | 0.0% | 1.8% | 0.0% | 0.00 | <0.1% | 1.9% | 0.0% | 0.00 |
| **F2040** | | | | | | | | | | |
| all Lats. | 302.7 | 15.2 | 3.3% | 2.3% | 0.3% | 0.05 | 6.7% | 1.8% | 0.8% | 0.11 |
| 30–35°N | 296.0 | 12.8 | 4.1% | 3.4% | 0.4% | 0.05 | 8.9% | 2.7% | 0.9% | 0.11 |
| 44–49°N | 313.6 | 19.1 | 1.3% | 1.2% | 0.2% | 0.03 | 3.8% | 0.8% | 0.5% | 0.10 |
| **F2090** | | | | | | | | | | |
| all Lats. | 321.1 | 17.0 | 2.7% | 2.6% | 0.1% | 0.02 | 7.3% | 2.1% | 0.4% | 0.07 |
| 30–35°N | 310.0 | 14.7 | 4.5% | 3.2% | 0.2% | 0.03 | 11.6% | 2.5% | 0.6% | 0.09 |
| 44–49°N | 334.3 | 21.5 | 0.2% | 1.9% | <0.1% | <0.01 | 0.8% | 1.8% | 0.1% | 0.02 |





tude regions (30–35°N and 44–49°N) are considered. Subsequently, the column ozone is calculated from the ozone profile. In Tab. 3 the total column ozone is shown as well as the ozone column in the mixing layer.

The ozone column in the mixing layer is assumed to correspond to the ozone column in a pressure range from 70–300 hPa. Despite the mixing layer comprises pressures between 70 and 300 hPa, not all air masses within this pressure range are necessarily

part of the mixing layer, because only air parcels above the thermal tropopause and with more than 31 ppbv CO are assumed to form the stratospheric mixing layer between tropospheric and stratospheric air. Hence, for determining the ozone column in the mixing layer, not only air masses in the mixing layer but also all further air masses between 70 and 300 hPa are considered. Since the composition of air parcels in the mixing layer consists of lowermost stratospheric air mixed with tropospheric air, the ozone mixing ratio in these air parcels is somewhat smaller than the mixing ratio in air masses with the same pressure

range and stratospheric character. Hence, the ozone column deduced from all air parcels in a pressure range from 70–300 hPa is expected to be somewhat larger than it would be considering only air parcels in the mixing layer. Thus in Tab. 3, the ozone column in the mixing layer might be overestimated.

In Sec. 4.3 the relative ozone destruction in the mixing layer caused by heterogeneous chlorine activation was determined and is given in Tab. 3. From the relative ozone loss and the ozone column in the mixing layer, the ozone loss caused by heterogeneous

chlorine activation in Dobson Units (DU) can be calculated.

The relative ozone loss in the mixing layer caused by heterogeneous chlorine activation is low. Thus, the maximum total ozone loss given in Tab. 3 is negligible compared with the total ozone column. Even in case F2040, where the chlorine activation causes most ozone destruction in the mixing layer, the total ozone loss accounts not to more than 0.05 DU. This are less than 0.1% of the total ozone column.

## 20   4.5   Likelihood of heterogeneous chlorine activation and its impact on ozone for low temperatures

As discussed in Sec 3.1, the temperatures in GLENS may be higher than in the real stratospheric mixing-layer. Therefore, a case study is performed assuming a shift in GLENS temperatures of –2 K to explore the impact of uncertainties in GLENS temperatures. The likelihood for the occurrence of heterogeneous chlorine activation assuming lower temperatures and its impact on ozone in the lowermost stratosphere is presented in Fig. 11.

The likelihood that chlorine activation occurs would increase significantly assuming lower temperatures (Fig. 11, top). Chlorine activation would occur with a likelihood of 3.7% for case C2010 instead of 1% assuming the temperatures from GLENS. Furthermore the likelihood would increase in the C2040 case. A fraction of 1.4% of all data in that case would cause chlorine activation (0.1% assuming GLENS temperatures). Applying geoengineering would cause the highest likelihood for chlorine activation to occur. In the case F2040 6.7% and in F2090 7.4% of the air masses would yield chlorine activation.

Despite of the higher likelihood of chlorine activation in the F2090 case, ozone is more affected in the F2040 case, because the ozone values in the range where ozone destruction would occur in the years 2040–2050 are higher than in the years 2090–2100 (not shown). Activated chlorine would destroy up to ∼0.8% of ozone in the lowermost stratosphere in the F2040 case, but only up to 0.4% in case F2090 (Fig. 11, bottom). Today and in the climate change scenario, more ozone would be likewise destroyed due to heterogeneous chlorine activation. This higher ozone destruction results in a reduced net ozone formation comparing





with the GLENS conditions. Except of the years 2090–2100 in the climate change scenario, the relative net ozone change (Fig. 11, middle) is significantly reduced for all cases considered. Comparing the behaviour in different latitude regions, the impact of heterogeneous chlorine activation on ozone is higher in lower latitudes.

The likelihood for heterogeneous chlorine activation to occur and its impact on ozone in the mixing layer determined in this section is summarized in Tab. 3 referred to as '2 K' lower temperatures. Assuming a reduction of 2 K of the GLENS temperatures increases the likelihood for heterogeneous chlorine activation to occur as well as its impact on lowermost stratospheric ozone. In all cases, the relative ozone loss in the mixing layer is two to three times higher assuming 2 K lower GLENS temperatures than in the reference. However, even for this upper limit, less than 0.8% of ozone in the mixing layer would be destroyed in the F2040 case, which shows the highest impact of activated chlorine on ozone. In all cases considered, an upper limit of

0.11 DU from a total ozone column of ∼303 DU in this region (which is less than 0.1%) has been estimated as the total ozone reduction caused by heterogeneous chlorine activation. Despite occurrence of heterogeneous chlorine activation, the net ozone change in the mixing layer during a 10 day period is found to be positive (hence, in all cases a net ozone formation occurs in the mixing layer).

## 5    Discussion

We analysed the relevance of heterogeneous chlorine activation for the ozone changes in the lowermost stratosphere today and in future assuming both climate change and the additional application of sulfate geoengineering.

Focussing on the GLENS mixing layer in mid extra-tropical latitudes, median ozone increases by 60 (30–35°N) –67% (44–49°N) by the end of the 21st century assuming climate change. In contrast, Ball et al. (2018) reported evidence for a decrease in mid-latitude lower stratospheric ozone between the years 1998 and 2016. This ozone decrease was attributed to be dynam-

ical driven (Chipperfield et al., 2018; Ball et al., 2019) by non-linear effects not yet completely understood and usually not implemented in climate models (Ball et al., 2019). However, this ozone decrease was found to be small with respect to the inter annual variability. This decrease results in a total ozone reduction of 1.9 DU between the years 1998 and 2018 in the lower stratosphere at 30–50°N. In comparison, ozone loss caused by heterogeneous chlorine activation analysed here and potentially occurring in the mid-latitude lower stratosphere in summer is found to cause an ozone loss of less than 0.1 DU for present day

conditions (case C2010) in the same latitude range.

This reduction of column ozone is determined here from the relative ozone loss in the GLENS mixing layer and the contribution of ozone in the mixing layer to total column ozone. In case C2010, GLENS air masses in a pressure range between 70 and 300 hPa contribute 4.9% to the ozone column in a latitude range of 30–35°N and 6.6% in 44–49°N. In comparison, Logan (1999) found a contribution of ozone between the thermal tropopause and 100 hPa on the total ozone column in summer of

∼6% in a latitude of 38°N and of ∼17% in 53°N. Thus in GLENS, the mixing layer contributes less to the ozone column than in the study of Logan (1999) deduced from satellite measurements between the years 1980 and 1993. However, if ozone in the GLENS mixing layer would contribute as much to column ozone as in the study of Logan (1999), only 0.03 DU of ozone would be destroyed for today's conditions at 30–35°N and 0.05 DU at 44–49°N.





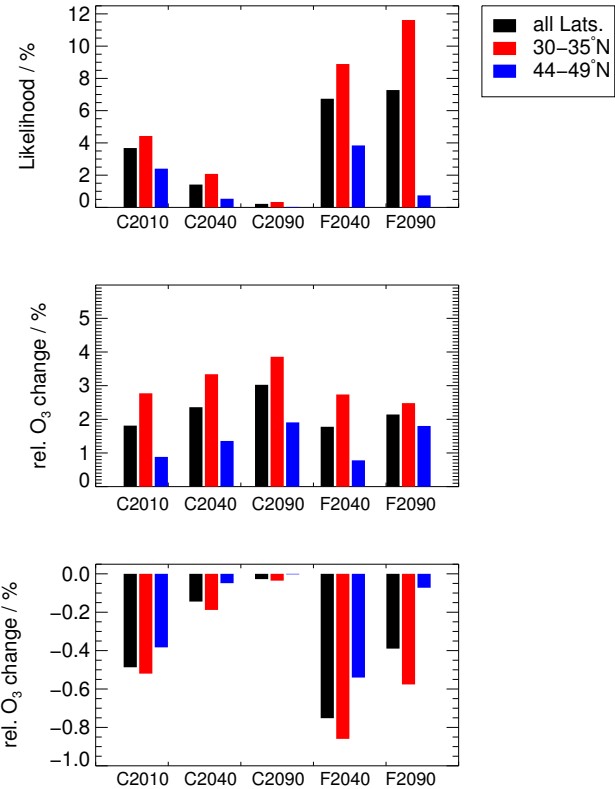

**Figure 11.** Likelihood (top) for the occurrence of chlorine activation as well as its impact on ozone in the lowermost stratosphere assuming 2 K lower temperatures than generated by GLENS. Further, the chemical ozone change in the mixing layer assuming 10 days without mixing of air parcels (middle) and the relative ozone change in the mixing layer caused by heterogeneous chlorine activation (bottom) is shown for the assumption with 2 K less temperatures. (See Tab. 1 for case descriptions.)

In all cases investigated here, a net chemical ozone formation occurs in the lowermost stratosphere. Ozone is formed there due to high CO and $CH_4$ mixing ratios which result from transport from the troposphere to the lowermost stratosphere. Hence, the oxidation of CO and $CH_4$, which usually forms ozone in the upper troposphere, causes ozone formation in the lowermost stratosphere as well (Lelieveld et al., 1997; Johnston and Kinnison, 1998). However, a potential ozone destruction in the mid-latitude lowermost stratosphere due to heterogeneous chlorine activation was discussed in previous studies (e.g. Keim et al., 1996; Anderson et al., 2012, 2017; Anderson and Clapp, 2018; Schwartz et al., 2013; Berthet et al., 2017; Robrecht et al., 2019; Clapp and Anderson, 2019; Schoeberl et al., 2020).

This chlorine driven ozone loss process could occur today above central North America in relation to stratospheric moistening through convective overshooting events during the North American Monsoon (NAM). Anderson et al. (2012) proposed a strong





impact of heterogeneous chlorine activation on ozone and Anderson and Clapp (2018) simulated a maximal fractional ozone loss of $-2.5$ to $-67\%$ (depending on the HCl mixing ratio) for the lower stratosphere between 12 km and 18 km assuming that conditions yielding heterogeneous chlorine activation as low temperatures and a high water vapour mixing ratio of 20 ppmv are maintained for 14 days. In contrast, Schwartz et al. (2013) argue that conditions cold and moist enough for chlorine acti-

vation are very rare and are usually associated with low HCl and ozone amounts. Schoeberl et al. (2020) found that lowermost stratospheric water vapour is increased during the NAM caused by enhanced convection followed by advection in the monsoon circulation. Simultaneously, the tropopause is uplifted reducing column ozone. However, this correlation between enhanced water vapour in the lowermost stratosphere and reduced column ozone is found to be dynamically driven with no evidence of substantial chemical ozone loss caused by chlorine activation. In our study, chlorine activation would reduce ozone in the

mixing layer by 0.1% (0.5% assuming 2 K lower temperatures). Nevertheless, ozone formation processes dominate ozone chemistry resulting in a net ozone formation simulated.

An enhancement of the stratospheric sulfate abundance, which causes ozone destruction in relation to heterogeneous chlorine activation, was mentioned in previous studies with respect to volcanic eruptions. Keim et al. (1996) combined laboratory measurements and observations and reported a removal layer for ozone caused by heterogeneous chlorine activation in the

mid-latitudes around the tropopause subsequent to the Mt. Pinatubo eruption. Solomon et al. (1998) accentuated the relevance of heterogeneous chlorine chemistry by calculating a column ozone loss of $\sim 4\%$ after the eruption of El Chicon and of $\sim 10\%$ after the eruption of Mt. Pinatubo in 40-50°N not considering fluctuations dynamical forcing between different years. However, (Solomon et al., 1998) considered the entire ozone column, not only ozone loss in the lowermost stratosphere.

In our study, only air masses are considered that are close to the tropopause. Nevertheless, it is very unlikely that ozone loss

caused by heterogeneous chlorine activation occurs at higher altitudes. As shown in section 4.2 the likelihood is highest for a pressure level around 140 hPa, because chlorine activation is favoured at high pressures. Furthermore, temperatures increase with altitude reducing the likelihood for chlorine activation to occur. Since the stratospheric $Cl_y$ concentration decreases because of the Montreal protocol, the impact of heterogeneous chlorine chemistry on ozone in the future is expected to be lower than for volcanic eruption in the past.

Comparing lowermost stratospheric ozone in the climate change and the geoengineering scenario, the overall ozone mixing ratio at the end of the 21st century is higher in the climate change scenario. There are other processes than heterogeneous chlorine activation, which are not investigated here and which would affect stratospheric ozone by applying sulfate geoengineering. For example, an increase in water vapour would increase $HO_x$ catalysed ozone destruction (Heckendorn et al., 2009), changes in radiation could affect oxygen and ozone photolysis. Furthermore, increased heterogeneous chemistry could enhance the

$NO_x$ concentration and gas phase chemistry could change due to higher stratospheric temperatures (Pitari et al., 2014). Since both chemistry and dynamics can affect stratospheric ozone at geoengineering conditions in multiple ways (e,g, Heckendorn et al., 2009; Pitari et al., 2014; Tilmes et al., 2009, 2014; Visioni et al., 2017b), further studies are necessary to assess the impact of geoengineering on stratospheric ozone (e.g. quantifying changes in $HO_x$ induced ozone destruction or investigating the dynamical contribution to the difference in the ozone mixing ratio between the climate change and the geoengineering

scenario).



# 6 Conclusions

Here, we focus on the potential occurrence of heterogeneous chlorine activation in the mixing layer between stratospheric and tropospheric air above central North America ($30.6 - 49.5°$N, $72.25 - 124.75°$W), which leads to catalytic ozone destruction known from the stratosphere during polar late winter and early spring. The likelihood for chlorine activation to occur and its

impact on ozone in the mixing layer today and in future is determined by comparing chlorine activation thresholds calculated based on CLaMS box-model simulations considering initial conditions for trace gases and aerosols from GLENS (*Geoengineering Large Ensemble simulations*) with the temperature and water vapour distribution in GLENS. In GLENS, two future scenarios are simulated with a global climate model from the years 2010–2100 considering both climate change following the RCP8.5 scenario and the additional application of geoengineering through stratospheric sulfate injections beginning in the year

2020 to keep the global mean temperature at the levels from 2020.

The GLENS mixing layer will warm and moisten in both future scenarios with a larger change in the geoengineering scenario (the median temperature is ∼2.5 K higher in the years 2090–2100 with geoengineering than with climate change and the median water vapour mixing ratio is ∼6.5 ppmv higher). The ozone mixing ratio increases in the mid-latitude mixing layer in GLENS assuming the climate change scenario, but is found to remain at today's level when sulfate geoengineering is applied.

These differences may be due to changes in both atmospheric dynamics and chemistry in the lowermost stratosphere. For example, potential chemical effects are an increasing $HO_x$-mixing ratio, because of a higher water vapour mixing ratio, or differences in the $NO_x$/$HNO_3$- or the $ClO_x$/$HCl$ partitioning driven by changes in the heterogeneous and gas phase chemistry. GLENS results in the mixing layer are analysed in comparison with SEAC[4]RS aircraft measurements. Most GLENS results and SEAC[4]RS measurements in the mixing layer range from 201–207 K and 5–8 ppmv $H_2O$. Thus, the water vapour and

temperature conditions in GLENS have a good overall agreement with current observations. However, in the SEAC[4]RS measurements a higher fraction of air parcels with very low temperatures of 197–200 K compared to GLENS results are found. Based on this difference catalytic ozone loss is additionally investigated for 2 K lower temperatures than in the GLENS results. A temperature and water vapour dependent threshold indicates conditions which lead to and which do not lead to heterogeneous chlorine activation. The chlorine activation threshold analysed in this study, marks an upper temperature limit for chlorine ac-

tivation to occur at a given water vapour mixing ratio. We showed that the chlorine activation thresholds depend on a variety of conditions. Increasing pressure, sulfate aerosol loading and ozone mixing ratio allow higher temperatures to cause chlorine activation. However, air parcels with higher pressures, sulfate aerosol loadings or ozone mixing ratios are usually warm. Hence, shifting chlorine activation thresholds to higher temperatures (and thus leading to a broader range of conditions allowing chlorine activation) does not necessarily increase the likelihood for chlorine activation to occur, because the temperatures of air

masses with the pressure and composition leading to these chlorine activation thresholds increase as well.

The likelihood for heterogeneous chlorine activation to occur and its impact on ozone in the mixing layer between tropospheric and stratospheric air masses is determined for several cases, which differ in the future scenario and in the considered years as further described in Tab. 1. The comparison of chlorine activation thresholds with GLENS results yields a likelihood for chlorine activation to occur of 1.0% for case C2010 and, assuming geoengineering, of 3.3% in case F2040 and 2.7% in case F2090.





In contrast, the likelihood is negligible in the climate change scenario (0.1% in case C2040 and 0.0% in C2090). Assuming 2 K lower temperatures, the likelihood increases accounting for 3.7% in case C2010, 6.7% and 7.3% in the cases F2040 and F2090, respectively, and 1.4% and 0.2% in the cases C2040 and C2090, respectively. We showed that the likelihood of occurrence is higher at lower latitudes and at higher pressure levels (lower altitudes). However, in air masses in which chlorine activation

may occur, usually a low ozone mixing ratio prevails. This fact contributes to the low impact of chlorine activation on ozone in the mixing layer.

The net chemical ozone change in the mixing layer is calculated here by combining the change in the ozone mixing ratio from 10 day CLaMS box-model simulations (final ozone − initial ozone) assuming specific water vapour and temperature conditions with the GLENS frequency distribution in the water vapour and temperature correlation. Normalizing this net chemical ozone

change with the total initial ozone in the mixing layer yields the net relative ozone change, which occurs if chemical processes proceed for 10 days without being perturbed by mixing between air parcels. Thus, in today's C2010 case the net relative ozone change in the mixing layer accounts for +2.3%. Also in the future scenarios, a net chemical ozone formation occurs. In the climate change future scenario, ozone increases within 10 days by ∼2.5% in case C2040 and by ∼3.0% in case C2090. In the sulfate geoengineering scenario, the increase is somewhat less with 2.3% in case F2040 and 2.6% in case F2090.

However, few ozone is destroyed caused by heterogeneous chlorine activation. The upper limit of ozone destruction in the stratospheric mixing layer caused by heterogeneous chlorine activation is 0.3% in the F2040 case, which is the case with the largest ozone destruction. Ozone destruction is larger in the subtropical latitude range (30–35°N). In that latitude range, 0.4% of ozone would be destroyed in in the F2040 case and 0.2% in 44–49°N. Assuming 2 K lower temperatures, less ozone would be formed during 10 days without mixing. Additionally the upper boundary for the relative ozone destruction in the mixing

layer above central North America would increase to 0.8% in the F2040 case.

Finally, the impact of heterogeneous chlorine activation in the mixing layer is estimated. Based on the conditions in GLENS, in all cases less than 0.1 DU of ozone (less than 0.1% of the ozone column) is destroyed caused by heterogeneous chlorine activation in the mixing layer. Assuming 2 K lower temperatures, not more than 0.11 DU of ozone are destroyed for all latitude regions and all cases today and in the future scenarios with climate change as well as with additional sulfate geoengineering.

In comparison in the Arctic polar winter in the year 2000, a volcanically clean year, 77±10 DU were destroyed between 110–30 hPa (Vogel et al., 2003) and (Tilmes et al., 2008) has shown significantly larger ozone depletion in high polar latitudes with geoengineering.

In summary we showed, that heterogeneous chlorine activation affects ozone in the lowermost stratosphere in mid-latitudes, but the impacts are very small. Sulfate geoengineering leads to a 2–3 times higher likelihood for the occurrence of chlorine

activation. However, in the geoengineering case most likely for chlorine activation, chlorine is activated with a probability of 3.3% (6.7% assuming 2 K lower temperatures) in the entire latitude region considered here. In all cases today and in future, less than 0.4% (0.9% assuming 2 K lower temperatures) of ozone in the mixing layer are destroyed caused by heterogeneous chlorine activation. We infer an upper limit for total ozone column reduction of 0.11DU (less than 0.1% of column ozone). Thus according to the results of this study, the relevance of ozone destruction caused by heterogeneous chlorine activation in





the mid-latitude mixing layer between stratospheric and tropospheric air is negligible with respect to the ozone column and small in the mixing layer even if sulfate geoengineering would be applied.

*Data availability.* Access options for GLENS data are given at https://doi.org/10.5065/D6JH3JXX. The results of CLaMS simulations can be requested from Sabine Robrecht (sa.robrecht@fz-juelich.de). The complete SEAC[4]RS data are availiabe at https://www-air.larc.nasa.gov/cgi-

bin/ArcView/seac4rs (last access: 27 May 2020, NASA, 2020) and options to acess START08 data are given at https://data.eol.ucar.edu/master_lists/genera (last access: 27 May 2020, NCAR, 2020).

## Appendix A: Comparison of the GLENS mixing layer with START08 measurements

In addition to measurements of the SEAC[4]RS campaign, data from the GLENS mixing layer in case C2010 are compared with measurements of the Stratosphere-Troposphere Analyses of Regional Transport (START08) aircraft campaign (Pan et al.,

2010). START08 aimed to investigate the stratosphere-to-troposphere transport focussing on stratospheric intrusions into the troposphere and tropospheric intrusions transporting air masses from the upper tropical troposphere to the extra-tropical low-ermost stratosphere. Flights during START08 took place from April – June 2008 and covered the area above central North America (25–65°N, 80–120°W) up to an altitude of ∼14.3 km (Pan et al., 2010).

In Fig. A1, the GLENS tracer-tracer-correlation of the GLENS mixing layer in case C2010 is compared with the mixing layer

deduced from START08 measurements. Air masses corresponding to the mixing layer between stratospheric and tropospheric air are assumed to be located above the thermal tropopause estimated based on the temperature–altitude profile during the flight. In addition they are selected to show more than 30 ppbv CO and more than 150 ppbv $O_3$. In contrast to the SEAC[4]RS mixing layer (Sec. 3.1), the ozone criteria is added to determine the mixing layer from START08 measurements, because many data points above the tropopause deduced from temperatures measured during the flight exhibited very low ozone mixing ratios

indicating a high fraction of air from the troposphere above the tropopause. This tropospheric air masses crossing the thermal tropopause were aimed to be probed during the START08 campaign.

The comparison of the stratospheric GLENS E90-$O_3$-correlation with the START08 CO-$O_3$-correlation is shown in black in Fig. A1 (top). In contrast to GLENS results, ozone measurements during START08 only reach up to ∼1400 ppbv $O_3$, because of the limitation of the probed altitude by the maximum flight height of ∼14.3 km. However, the mixing layer in GLENS (red)

comprises a similar ozone range as the mixing layer deduced from START08 measurements.

In Fig A1 (middle), the $H_2O$-temperature correlation (left) and the $H_2O$-$O_3$ correlation (right) for the entire latitude region considered in this study is shown. Fig A1 (bottom) shows the same correlations for the mixing layer deduced from START08 measurements. The water vapour mixing ratio exceeds in both model results and measurements more than 30 ppmv and the ozone mixing ratio ranges mainly between 300 and 500 ppbv. However, GLENS temperatures are somewhat lower than tem-

peratures measured during START08.



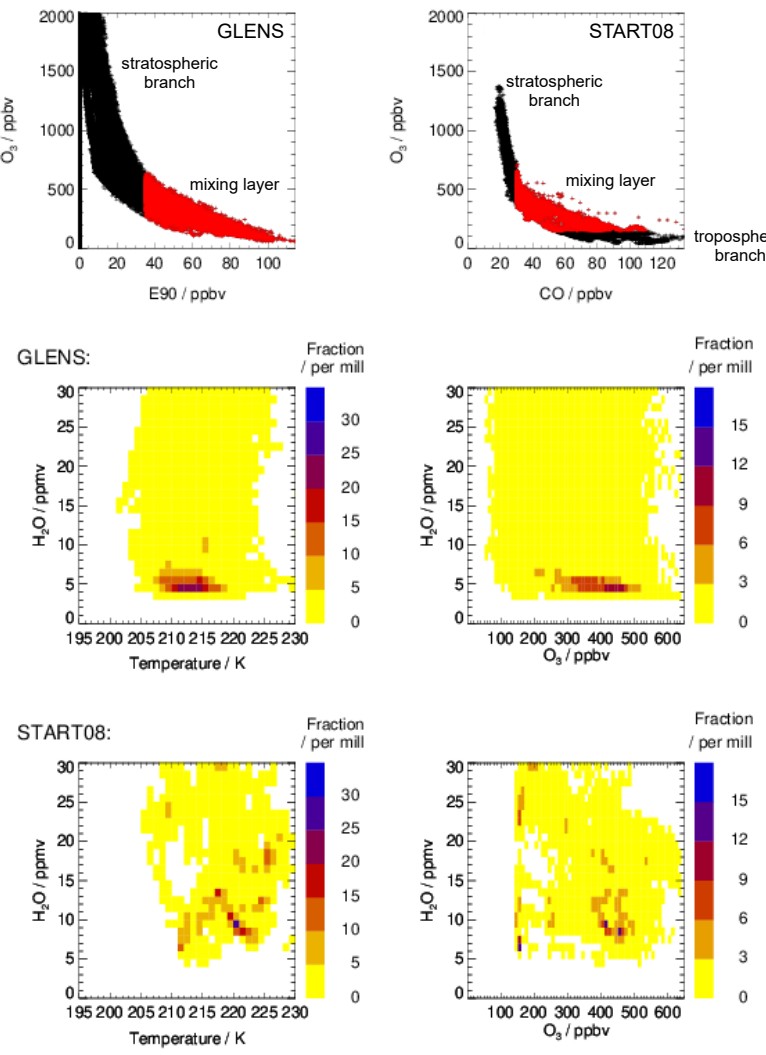

**Figure A1.** Comparison of the GLENS mixing layer between stratospheric and tropospheric air masses of the C2010 case with measurements of the START08 aircraft campaign. Top panels show the E90-$O_3$ correlation of GLENS air masses (left) and the CO-$O_3$-relative frequency distribution of START08 measurements (right) for stratospheric air masses (black) and air masses corresponding to the mixing layer (red). The middle panels show the relative frequency distribution in the $H_2O$-Temperature (left) and the $H_2O$-$O_3$ (right) correlation of the GLENS mixing layer and bottom panels of the mixing layer deduced from START08 measurements.



*Author contributions.* All authors developed the concept of the study. SR conducted CLaMS simulations, analysed GLENS results and wrote the manuscript with contributions from all authors.

*Competing interests.* The authors declare, that they have no conflict of interests.

*Acknowledgements.* Our activities were funded by the German Science Foundation (Deutsche Forschungsgemeinschaft, DFG) under the
5  DFG project CE-O$_3$ in the context of the Priority Program Climate Engineering: Risks, Challenges, Opportunities? (SPP 1689; VO 1276/4-1). We thank the groups of Ru-Shan Gao (NOAA Chemical Sciences Laboratory, Boulder, CO, USA), Jessica Smith (Havard University, Department Earth and Planetary Science, Cambridge, MA, USA) and Steven Wofsy (Havard University, Department Earth and Planetary Science, Cambridge, MA, USA) for providing their data measured during the SEAC$^4$RS aircraft campaign. We further thank Dale Hurst (NOAA ESRL Global Monitoring Division, Boulder, CO, USA) and the NCAR Earth Observating Laboratory team (NCAR, Boulder, CO, 
10  USA) for providing START08 data. For GLENS, computing resources were provided by the Climate Simulations Laboratory (CISL).





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
