# Peer review of "Potential of future stratospheric ozone loss in the mid-latitudes under global warming and sulfate geoengineering"

_Atmospheric Chemistry and Physics, 2020_

## Referee Comment (RC1) · Daniele Visioni (Referee) · 27 Aug 2020

This study by Robrecht et al. focuses on analysing the conditions over which heterogeneous chlorine activation and its subsequent effect on ozone concentration in the mid-latitudes might happen if stratospheric aerosols injections are applied. To do so, the authors use both direct simulations results from CESM1(WACCM) and box-model simulations from the Chemical Lagrangian Model of the Stratosphere (CLaMS). While the impact is found to be rather minimal, the analyses presented in this paper are very nicely described, are scientifically robust and definitely of interest to the scientific community. The paper is of high quality and deserves publication on ACP. I have some

minor suggestions that I list below, and after they are addressed the manuscript can be accepted promptly.

General remark: Throughout the entire manuscript, the authors define the RCP8.5 cases as the "climate change" cases, whereas the GLENS geoengineering scenario is defined as "geoengineering scenario". They also are not always consistent with the names they use to define the two scenarios. I think that's not entirely correct, as a definition, and could be a bit misleading. Would climate change under RCP8.5? Of course, as a consequence of high GHGs mixing ratios. But contrasting "climate change" with geoengineering gives the impression that what GLENS (and geoengineering in general) does is "cancels out" climate change. It doesn't, and plenty of GLENS studies have shown that, while smaller than those produced by the warming alone, geoengineering does produce some changes in the surface climate. So, for clarity, I would suggest the authors use a more precise definition for the RCP8.5 scenario (i.e. "increased surface warming"). Even just calling it "RCP8.5" would be better.

Page 1 Line 3: no comma after "Especially" Line 16: comma after furthermore Line 17: "has been" discussed? Line 29: I would suggest also citing some of the results from CCMI, for instance discussing the sensitivity of ozone changes to various ODSs and GHGs, Morgenstern et al. (2018)

Page 4 Line 22: focusing with just one s

Page 5 Line 23: maybe better to specify it's "surface" temperatures that GLENS tries to manage? Line 30: would be emitted by the end Line 31: in CESM1(WACCM), yes, but other models need to inject upward to 20 Tg-SO2 to get the proper AOD (see Timmreck et al. (2018) and references therein). So either specify that that's what WACCM needs, or that there's a range of uncertainty in this.

Page 6 Table 1: I think the names of the scenarios should be more consistent between experiments, as I said before. First of all, the 2010-2020 period is still under RCP8.5, since the emissions do vary even in that decade between the RCPs. Second, if C2040
is defined as "Climate Change following the RCP8.5 emission pathway" (although here I would use "surface temperatures increase"), then consistently F2040 should be defined as "Sulfate geoengineering to maintain temperatures at C2010 levels" (but it's actually 2010-2030, so it should be more specific) and similarly for F2090. Or another idea could be to have a column for "Underlying Emission Scenario" (that is always RCP8.5), a column for "Global T increase" (that would be > 0 for the C simulations and ~0 for the F simulations) and a column for "Stratospheric SO2 injected" (that would be = 0 for the C simulations and »0 for the F simulations). This would give the reader unfamiliar with GLENS an estimate at a glance of the amount of warming and of intervention that it's being considered here. Overall, the authors are free to do as they wish, but I strongly suggest making this table more useful to the reader.

Line 15 or thereabout: is the spatial range chosen for a specific purpose (i.e. we have measurements over that area) or just as an example because (clearly) considering the entire latitudinal band would be too much calculations? From reading further on, it is clear it is the former, so it should be specified in here too.

Page 7 Line 3: no "in" before "the focus of this study"

Page 8 Table 2: "For a better overview in this paper, the pressure ranges are allocated to a pressure level" I find this phrase very hard to parse, even if I can catch the meaning. Should be rewritten. Line 5: "tropospheric character" doesn't really mean much. Maybe "characteristic"?

Page 12 Lines 2-5: If the "higher ozone smog production" is a result of RCP8.5 emissions, shouldn't the same also be observable in GLENS, given the underlying emissions are the same? Later, on line 5, should specify that the BDC changes transporting more air downward are true at the considered latitudes, not everywhere.

Page 14 Line 17: no comma after "range"

Page 18 Line 1-2: the HCl absorption by the liquid aerosols is an interesting point.
Some mention of the fact that this has been observed and discussed for volcanic eruption could be of interest to the reader (see Tabazadeh and Turco, 1993) to back up this point.

Page 21 Line 1: "at" lower latitudes than "at" higher...

Line 15-16: The use of commas in this phrase makes it a bit hard to follow: I suggest a slight change below "Since no mixing is allowed in the box-model runs, the conditions that yield chlorine activation are not disturbed within 10 days"

Line 17: maybe "conservation" is a clearer term than "maintenance"?

Line 34: this makes it sound like the geoengineering scenario in GLENS prescribes a reduction of ODS. That reduction is prescribed in the underlying RCP scenarios, more in general. This is especially confusing considering the authors sometimes use GLENS to identify the geoengineering scenario, sometimes both future scenarios.

Page 23 Line 26: "are" summarized

Page 25 Line 6: are reduced "by" 2K

Page 26 Line 4: "even though" is better here, otherwise "despite the fact that"

Line 30: "Despite the"

Page 27 Line 1: Except "for"

Throughout the Discussion session, is hard to understand what the authors mean by GLENS. The geoengineering scenario? The RCP8.5 one? Should be clarified.

Page 29 Line 19: "In our study, only air masses that are close to the tropopause are considered"

Page 31 Line 15: "few ozone" is not really correct. "Not much", or "scarcely any" would me more correct. Also, "due to" (or "as a consequence of") instead of "caused by" Line 28: no comma after showed

**ACPD**
References

Morgenstern, O., Stone, K. A., Schofield, R., Akiyoshi, H., Yamashita, Y., Kinnison, D. E., Garcia, R. R., Sudo, K., Plummer, D. A., Scinocca, J., Oman, L. D., Manyin, M. E., Zeng, G., Rozanov, E., Stenke, A., Revell, L. E., Pitari, G., Mancini, E., Di Genova, G., Visioni, D., Dhomse, S. S., and Chipperfield, M. P.: Ozone sensitivity to varying greenhouse gases and ozone-depleting substances in CCMI-1 simulations, Atmos. Chem. Phys., 18, 1091–1114, https://doi.org/10.5194/acp-18-1091-2018, 2018.

Tabazadeh, A., & Turco, R. (1993). Stratospheric Chlorine Injection by Volcanic Eruptions: HCl Scavenging and Implications for Ozone. Science, 260(5111), 1082-1086. Retrieved August 19, 2020, from www.jstor.org/stable/2881602

Timmreck, C., Mann, G. W., Aquila, V., Hommel, R., Lee, L. A., Schmidt, A., Brühl, C., Carn, S., Chin, M., Dhomse, S. S., Diehl, T., English, J. M., Mills, M. J., Neely, R., Sheng, J., Toohey, M., and Weisenstein, D.: The Interactive Stratospheric Aerosol Model Intercomparison Project (ISA-MIP): motivation and experimental design, Geosci. Model Dev., 11, 2581–2608, https://doi.org/10.5194/gmd-11-2581-2018, 2018.

---

## Referee Comment (RC2) · Anonymous Referee #2 · 14 Sep 2020

This is a well-designed study that seeks to examine the potential for and impacts of heterogeneous chlorine activation in the lower stratosphere on ozone in current and future climates. In particular, using global climate model projections of the future with and without geoengineering assumptions, the likelihood of chlorine activation is assessed over time and evaluated to assess future impacts on lower stratosphere ozone. Overall, I found the paper to be a valuable contribution and worthy of publication. My only significant criticism would be in the weight given to the results and their interpretation in the text throughout, as the narrative broadly glances over the limitations/caveats of the GLENS model that are relevant to the subject matter. This is not to say that important elements are ignored or simply not acknowledged. Rather, they are largely

dismissed when discussing the significance of the results. Either more evidence needs to be given to favor or increase confidence the GLENS results or the limitations of the model should be more routinely stated in context of the results.

**Major points:** One of the more concerning limitations of GLENS in my assessment is the apparent warm bias of the model in the UTLS (which is common in most models given their relatively coarse vertical resolution near the tropopause). In the paper, this is assessed using airborne observations from the SEAC4RS campaign, which are very good but ultimately too limited for comprehensive validation of the model. I would strongly recommend that the authors consider using high-resolution radiosonde observations to characterize the true temperature bias in the UTLS (by comparing tropopause-relative T) as it may be as high as 5 K based on the data shown and is a major source of sensitivity to the chlorine activation results. The authors do show what an assumed warm bias of 2 K would lead to, but even this number appears to be conservative in my opinion. Rather than the messaging throughout stating that the assessments in the paper are an "upper bound" to chlorine activation, I would argue that in many ways they are a lower bound. Better assessments of model biases will help to focus the messaging more on the expected likelihood and impacts of this important process.

The second limitation that I believe needs to be better addressed and highlighted is the representation of convection in the GLENS model. Climate models are often not classified as resolving (or even representing) convection well. Rather, global coarse horizontal resolution models such as GLENS are often better used to assess changes in convective environments. Dynamically downscaled climate simulations have become increasing used to study convection since it can be better simulated (and even resolved) over regions of interest by using the large-scale environments projected by the global model as input. Since this study relies on the global large-scale climate projection alone, the realism of UTLS water vapor and its variability due to convection is highly questionable. It is very likely a significantly underestimated reference point,

which again is in contrast to the messaging throughout in the paper. I would like to see these points better highlighted and used to interpret the results. I'm not asking for additional analysis to respond to this point, but more appropriate messaging/discussion in the text and perceived importance or likelihood of chlorine activation.

Apart from these points on the under-emphasized model limitations, I don't have an exhaustive list of technical corrections - the text and figures are otherwise excellent.

**Minor points:** Page 11, line 4 - should cite Smith et al, 2017 (doi:10.1002/2017JD026831) and Herman et al, 2017 (doi:10.5194/acp-17-6113-2017) as well since this studies more extensively evaluate delivery of water to the stratosphere by convection during SEAC4RS.

Page 26, line 21 - contrary to this statement, I found very little discussion of the apparent temperature bias in GLENS in Section 3.1.

───────────────────

---

## Author Comment (AC1) · 7 Dec 2020

**Author Comment to Referee #1 (Daniele Visioni)**

ACP Discussions doi:10.5194/acp-2020-747 (Editor - Farahnaz Khosrawi) 'Potential of future stratospheric ozone loss in the midlatitudes under climate change and sulfate geoengineering'

We thank Daniele Visioni for detailed guidance on how to revise our paper. Following the reviewers advice we renamed the "climate change" scenario to "global warming" scenario. Further, we took more care by using the term "GLENS" in the discussion and specified the future scenarios (geoengineering and global warming) more precisely. Our reply to the reviewer comments is listed in detail below. Questions and comments of the referee are shown in italics. Passages from the revised version of the manuscript are shown in blue. Comments of the reviewer, which are not listed below, were applied to the manuscript as the reviewer suggested. In general, comments not shown here refer to typos and grammar mistakes.

**Referee Comments**

This study by Robrecht et al. focuses on analysing the conditions over which heterogeneous chlorine activation and its subsequent effect on ozone concentration in the mid-latitudes might happen if stratospheric aerosols injections are applied. To do so, the authors use both direct simulations results from CESM1(WACCM) and box-model simulations from the Chemical Lagrangian Model of the Stratosphere (CLaMS). While the impact is found to be rather minimal, the analyses presented in this paper are very nicely described, are scientifically robust and definitely of interest to the scientific community. The paper is of high quality and deserves publication on ACP. I have some minor suggestions that I list below, and after they are addressed the manuscript can be accepted promptly.

General remark: Throughout the entire manuscript, the authors define the

RCP8.5 cases as the "climate change" cases, whereas the GLENS geoengineering scenario is defined as "geoengineering scenario". They also are not always consistent with the names they use to define the two scenarios. I think that's not entirely correct, as a definition, and could be a bit misleading. Would climate change under RCP8.5? Of course, as a consequence of high GHGs mixing ratios. But contrasting "climate change" with geoengineering gives the impression that what GLENS (and geoengineering in general) does is "cancels out" climate change. It doesn't, and plenty of GLENS studies have shown that, while smaller than those produced by the warming alone, geoengineering does produce some changes in the surface climate. So, for clarity, I would suggest the authors use a more precise definition for the RCP8.5 scenario (i.e. "increased surface warming"). Even just calling it "RCP8.5" would be better.

We changed the name of the "climate change" scenario to "global warming" throughout the paper. Further, the terms for both GLENS scenarios are used more precisely in the paper and are more specifically introduced (p.6, 1.5–9 of the revised manuscript):

GLENS provides a comprehensive global data set assuming two different potential scenarios and covering the years 2010–2100. The GLENS scenario, which follows the RCP8.5 emission pathway, will lead to an increased warming of the global mean surface temperature in future. Hence, this scenario is referred to as the "global warming scenario" in this study. The GLENS future scenario, which assumes the RCP8.5 emission pathway together with stratospheric  $SO_2$  injections to keep the global mean surface temperature from warming, is here referred to as the "geoengineering scenario".

Page 1 Line 29: I would suggest also citing some of the results from CCMI, for instance discussing the sensitivity of ozone changes to various ODSs and GHGs, Morgenstern et al. (2018)

We added 2 sentences with the results of Morgenstern et al. (2018) at p. 1 l. 28–32 of the revised manuscript:

However, comparing simulations of different models Morgenstern et al. (2018) show that an increase in CH4 can also lead to an ozone reduction in the lowermost stratosphere. Increasing N2O mixing ratios lead to an increase in ozone for most model simulations Morgenstern et al. (2018) compared, while more  $CO_2$  likely causes an ozone reduction in the tropical and subtropical lowermost stratosphere.

Page 5 Line 23: maybe better to specify it's "surface" temperatures that GLENS tries to manage?

We agree with the reviewer and clarify that the temperature targets in the GLENS geoengineering scenario are for the surface temperature (p. 5, l. 25–27 of the revised manuscript).

The geoengineering scenario of GLENS is based on the RCP8.5 scenario, but aims to hold the global mean temperature, the inter-hemispheric temperature gradient and the equator-to-pole gradient at the Earth surface at the level to the year 2020 by applying stratospheric sulfur injections (for more details see Kravitz et al. (2017)).

Page 5 Line 31: in CESM1(WACCM), yes, but other models need to inject upward to 20 Tg-SO2 to get the proper AOD (see Timmreck et al. (2018) and references therein). So either specify that that's what WACCM needs, or that there's a range of uncertainty in this.

We adapted that sequence in the revised manuscript (p. 5, l. 32 - p. 6, l. 4):

To reach the temperature targets, in the GLENS geoengineering scenario more than 50 Tg SO2 would have been emitted at the end of the 21st century. This is five times the emitted amount of sulfur by the Mt. Pinatubo eruption in the year 1992 (Tilmes et al., 2018). However, other models than the WACCM do need to inject other amounts of SO2 into the stratosphere to keep the global mean surface temperature constant (Timmreck et al., 2018). Hence, it should be noted that generally there is a certain range of uncertainty in the SO2 amount needed.

Page 6 Table 1: I think the names of the scenarios should be more consistent

between experiments, as I said before. First of all, the 2010-2020 period is still under RCP8.5, since the emissions do vary even in that decade between the RCPs. Second, if C2040 is defined as "Climate Change following the RCP8.5 emission pathway" (although here I would use "surface temperatures increase"), then consistently F2040 should be defined as "Sulfate geoengineering to maintain temperatures at C2010 levels" (but it's actually 2010-2030, so it should be more specific) and similarly for F2090. Or another idea could be to have a column for "Underlying Emission Scenario" (that is always RCP8.5), a column for "Global T increase" (that would be > 0 for the C simulations and ~0 for the F simulations) and a column for "Stratospheric SO2 injected" (that would be = 0 for the C simulations and >0 for the F simulations). This would give the reader unfamiliar with GLENS an estimate at a glance of the amount of warming and of intervention that it's being considered here. Overall, the authors are free to do as they wish, but I strongly suggest making this table more useful to the reader.

We thank the reviewer for the advise and extended Tab. 1 in the manuscript. Further, we declared the cases more specific and consistent throughout the paper.

Table 1: Overview of the cases analysed in this study. In addition to the years considered, the underlying emission scenario in the GLENS simulation, the global temperature increase (referred to 2010-2020) and the SO2-amount injected by that time period are given for each case.

| Case  | Years     | GLENS scenario | emission sce- | global tempera-   | ${f SO}_2 {f injected} /$ |
|-------|-----------|----------------|---------------|-------------------|---------------------------|
|       |           |                | nario         | ture increase / K | $\mathbf{Tg}$             |
| C2010 | 2010-2020 | global warming | RCP8.5        | 0.0               | 0.0                       |
| C2040 | 2040-2050 | global warming | RCP8.5        | 1.8               | 0.0                       |
| C2090 | 2090-2100 | global warming | RCP8.5        | 6.0               | 0.0                       |
| F2040 | 2040-2050 | geoengineering | RCP8.5        | -0.1              | 14.4                      |
| F2090 | 2090-2100 | geoengineering | RCP8.5        | 0.1               | 49.0                      |

The cases are introduced at p. 6, l. 12–17 of the revised version of the manuscript:

In total 5 cases are analysed which are determined through the GLENS scenario and the decade. The case C2010 describes conditions in the early 21st century (2010–2020) based on the GLENS global warming scenario.

The conditions for the mid (2040-2050) and the end (2090-2100) of the 21st century following the global warming scenario are referred to as case C2040 and C2090, respectively. The cases of the geoengineering scenario are named F2040 and F2090 for the mid and the end of the 21st century, respectively ("F" stands for the "Feedback" mechanism of the SO2 injections). An overview of the considered cases is given in Tab. 1 together with the global mean temperature increase reached in that case comparing to the conditions of the years 2010–2020 and the injected amount of SO2.

Page 6 Line 15 or thereabout: is the spatial range chosen for a specific purpose (i.e. we have measurements over that area) or just as an example because (clearly) considering the entire latitudinal band would be too much calculations? From reading further on, it is clear it is the former, so it should be specified in here too.

We specified the purpose for the selection of the considered area (p. 6, l. 18–20 of the revised version of the manuscript):

GLENS results are selected for a latitude range of  $30.6-49.5^{\circ}$ N and a longitude range of  $72.25-124.75^{\circ}$ W (grey marked in Fig. 1, left), because for this area the reliability of the GLENS C2010 results could be analysed in comparison with aircraft measurements of the SEAC4RS and START08 campaigns.

Page 8 Table 2: "For a better overview in this paper, the pressure ranges are allocated to a pressure level" I find this phrase very hard to parse, even if I can catch the meaning. Should be rewritten.

We rewrote the sentence in the caption of Tab. 2 (p. 9, l. 2–3 in the revised version of the manuscript).

For a better overview in this paper, pressure levels are used to describe the pressure ranges (e.g. 80 hPa level for the pressure range 70–90 hPa).

Page 8, Line 5: "tropospheric character" doesn't really mean much. Maybe "characteristic"?

We changed the sentence as follows (p.8, l.1–2 of the revised version of the manuscript):

Furthermore, based on the ozone mixing ratio considered air masses can be divided in those with a chemical composition of air masses typically for the troposphere (low ozone) and those with a chemical composition more typically for the stratosphere (high ozone).

Page 12 Lines 2-5: If the "higher ozone smog production" is a result of RCP8.5 emissions, shouldn't the same also be observable in GLENS, given the underlying emissions are the same? Later, on line 5, should specify that the BDC changes transporting more air downward are true at the considered latitudes, not everywhere.

We agree with the reviewer that a higher ozone smog formation would be also expected in the GLENS geoengineering scenario, because the emissions of  $CO_2$  and  $CH_4$  are the same as in the RCP8.5 scenario. However, we did not find an obvious reason, for the lower ozone mixing ratios in the geoengineering cases. Because of more extensive research on this question is not in the scope of our study, we removed that sentence from the paper.

Regarding the changes in the Brewer-Dobson-Circulation (BDC), we added that the statement refers to the considered latitude range (p. 12, l. 12–16 of the revised version of the manuscript).

For cases with global warming, the ozone mixing ratio is significantly higher in case C2040 (yellow) and C2090 (green) especially for low E90 concentrations. The enhancement of ozone in the mixing layer could be caused by changes in atmospheric transport or chemistry. Global warming is expected to increase upper stratospheric ozone and accelerate the BDC. In the considered latitude range, this leads to more ozone transported downwards in the lowermost stratosphere from high altitudes (Iglesias-Suarez et al., 2016).

Page 18 Line 1-2: the HCl absorption by the liquid aerosols is an interesting point. Some mention of the fact that this has been observed and discussed for volcanic eruption could be of interest to the reader (see Tabazadeh and Turco, 1993) to back up this point. We thank the referee for recommending us to cite Tabazadeh and Turco (1993). We mentioned the results of that study (p. 17, l. 21–23 in the revised version of the manuscript):

HCl uptake into supercooled water particles was also found to occur after volcanic eruptions resulting in an "HCl scavenging", which may protect the ozone layer (Tabazadeh and Turco, 1993).

Page 21 Line 17: maybe "conservation" is a clearer term than "maintenance"?

We substituted the term maintenance with the term conservation (p. 21, l. 26–27 in the revised version of the manuscript).

In the lowermost stratosphere, the duration of conservation for conditions causing chlorine activation is not yet known.

Line 34: this makes it sound like the geoengineering scenario in GLENS prescribes a reduction of ODS. That reduction is prescribed in the underlying RCP scenarios, more in general. This is especially confusing considering the authors sometimes use GLENS to identify the geoengineering scenario, sometimes both future scenarios.

Throughout the paper, we took care to specify more precise the discussed GLENS scenario and the considered case. Here, we adjusted the sequence to make clear that ODS are reduced in both GLENS scenarios in the future (p. 22, l. 10–11 of the revised version of the manuscript).

The increasing ozone formation in the future may be related to the reduction of ODS implemented in both GLENS scenarios.

Throughout the Discussion session, is hard to understand what the authors mean by GLENS. The geoengineering scenario? The RCP8.5 one? Should be clarified.

We thank the referee for this remark. As said before, throughout the manuscript we took care to specify more precise the discussed GLENS scenario and the considered case.

Page 31 Line 15: "few ozone" is not really correct. "Not much", or "scarcely any" would be more correct.

Because we did not analyse the temperature bias of GLENS extensively, it might be possible that the temperatures assumed in our study are to high. Performing case studies with lower temperatures yields that lower temperatures would increase the potential impact of heterogeneous chlorine activation on ozone. Because of this range of uncertainty, we decided to keep "few ozone" here.

**References**

- Iglesias-Suarez, F., Young, P. J., and Wild, O.: Stratospheric ozone change and related climate impacts over 1850–2100 as modelled by the ACCMIP ensemble, Atmos. Chem. Phys., 16, 343–363, doi:10.5194/acp-16-343-2016, 2016.
- Kravitz, B., MacMartin, D. G., Mills, M. J., Richter, J. H., Tilmes, S., Lamarque, J.-F., Tribbia, J. J., and Vitt, F.: First simulations of designing stratospheric sulfate aerosol geoengineering to meet multiple simultaneous climate objectives, J. Geophys. Res., 122, 12616–12634, doi: 10.1002/2017JD026874, 2017.
- Morgenstern, O., Stone, K. A., Schofield, R., Akiyoshi, H., Yamashita, Y., Kinnison, D. E., Garcia, R. R., Sudo, K., Plummer, D. A., Scinocca, J., Oman, L. D., Manyin, M. E., Zeng, G., Rozanov, E., Stenke, A., Revell, L. E., Pitari, G., Mancini, E., Di Genova, G., Visioni, D., Dhomse, S. S., and Chipperfield, M. P.: Ozone sensitivity to varying greenhouse gases and ozone-depleting substances in CCMI-1 simulations, Atmos. Chem. Phys., 18, 1091–1114, doi:10.5194/acp-18-1091-2018, 2018.

Tabazadeh, A. and Turco, R. P.: Stratospheric Chlorine Injection by Volcanic Eruptions: HCl Scavenging and Implications for Ozone, Science, 260, 1082–1086, doi:10.1126/science.260.5111.1082, 1993.

- Tilmes, S., Richter, J. H., Kravitz, B., MacMartin, D. G., Mills, M. J., Simpson, I. R., Glanville, A. S., Fasullo, J. T., Phillips, A. S., Lamarque, J.-F., et al.: CESM1 (WACCM) stratospheric aerosol geoengineering large ensemble project, Bull. Am. Meteorol. Soc., 99, 2361–2371, doi:10.1175/ BAMS-D-17-0267.1, 2018.
- Timmreck, C., Mann, G. W., Aquila, V., Hommel, R., Lee, L. A., Schmidt, A., Brühl, C., Carn, S., Chin, M., Dhomse, S. S., Diehl, T., English, J. M., Mills, M. J., Neely, R., Sheng, J., Toohey, M., and Weisenstein, D.: The Interactive Stratospheric Aerosol Model Intercomparison Project (ISA-MIP): motivation and experimental design, Geosci. Model Dev., 11, 2581–2608, doi:10.5194/gmd-11-2581-2018, 2018.

---

## Author Comment (AC2) · 7 Dec 2020

**Author Comment to Referee #2**

ACP Discussions doi:10.5194/acp-2020-747 (Editor - Farahnaz Khosrawi) 'Potential of future stratospheric ozone loss in the midlatitudes under climate change and sulfate geoengineering'

We thank referee #2 for specific guidance on how to revise our paper. We performed a sensitivity study assuming 5 K less temperatures than simulated in GLENS to show the impact of lower temperatures for this ozone loss process more clearly. Further, we discussed the missing of convective overshooting events shortly and do not claim our results "upper boundary" any more. Our reply to the reviewer comments is listed in detail below. Questions and comments of the referee are shown in italics. Passages from the revised version of the manuscript are shown in blue.

This is a well-designed study that seeks to examine the potential for and impacts of heterogeneous chlorine activation in the lower stratosphere on ozone in current and future climates. In particular, using global climate model projections of the future with and without geoengineering assumptions, the likelihood of chlorine activation is assessed over time and evaluated to assess future impacts on lower stratosphere ozone. Overall, I found the paper to be a valuable contribution and worthy of publication. My only significant criticism would be in the weight given to the results and their interpretation in the text throughout, as the narrative broadly glances over the limitations/caveats of the GLENS model that are relevant to the subject matter. This is not to say that important elements are ignored or simply not acknowledged. Rather, they are largely dismissed when discussing the significance of the results. Either more evidence needs to be given to favor or increase confidence the GLENS results or the limitations of the model should be more routinely stated in context of the results.

**Major points**

One of the more concerning limitations of GLENS in my assessment is the apparent warm bias of the model in the UTLS (which is common in most models given their relatively coarse vertical resolution near the tropopause). In the paper, this is assessed using airborne observations from the SEAC4RScampaign, which are very good but ultimately too limited for comprehensive validation of the model. I would strongly recommend that the authors consider using high-resolution radiosonde observations to characterize the true temperature bias in the UTLS (by comparing trop pause-relative T) as it may be as high as 5 K based on the data shown and is a major source of sensitivity to the chlorine activation results. The authors do show what an assumed warm bias of 2 K would lead to, but even this number appears to be conservative in my opinion. Rather than the messaging throughout stating that the assessments in the paper are an "upper bound" to chlorine activation, I would argue that in many ways they are a lower bound. Better assessments of model biases will help to focus the messaging more on the expected likelihood and impacts of this important process.

We agree with the referee that the warm bias of the model could be as high as 5 K from the comparison with SEAC4RS measurements. Determining the temperature bias more extensively (e.g. using radiosonde observations) would be enough content of a new study. Hence, it is out of the scope of our study, which focuses on the likelihood for the ozone loss process to occur in the mid-latitude lowermost stratosphere. Because the temperature plays a key role for the likelihood of the ozone loss process, we analysed a further sensitivity case with 5 K lower temperatures in Sec. 4.5 of the revised version of the manuscript. The results of this case study are also included in the conclusions to estimate the range of uncertainty of our results better.

The occurrence of conditions which would lead to chlorine activation and thus ozone loss in the mid-latitude lowermost stratosphere is not yet clear (e.g. the occurrence frequency of convective overshooting, the temperature and the ozone mixing ratio of moist air masses, the duration of the conservation of low temperatures together with elevated water vapour). Because of this range of uncertainty, we decided to claim our results not as "upper boundary" to the impact of chlorine activation on ozone.

Sec. 4.5 was revised as follows discussing in addition a temperature shift of

-5K.

**Likelihood of heterogeneous chlorine activation and its impact on ozone for low temperatures**

As analysed in Sec. 3.1, the temperatures in the mixing-layer above the tropopause simulated in GLENS may be higher than the real atmospheric temperatures in this region. Therefore, a sensitivity study is performed assuming a shift in GLENS temperatures of -2 K and of -5 K to explore the impact of uncertainties in the temperatures calculated in GLENS. However, the focus of this sensitivity assumption is only on the temperature shift without considering a potential ice formation at very low temperatures. The likelihood for the occurrence of heterogeneous chlorine activation assuming lower temperatures and its impact on ozone in the lowermost stratosphere is presented in Fig. 1 (of this reply).

[revised manuscript text omitted]

---

## Editor Decision (ED1)

**Technical corrections on Manuscript No. ACP-2020-747 (Robrecht et al.)**

P1, L12: add "model" after GLENS

P2, L11: space between number in percent and latitude in parentheses missing

P2, L31: Something is wrong in this sentence. Please check and correct.

P5, L14: I would use here singular, thus "temperature", "atmospheric wind".

P5, L16: skip "s" and write just "Mt. Pinatubo eruption".

P6, L21: acronyms should be introduced.

P9, L4: thresholds -> threshold

P9, L17: write H2O instead of water vapour. This is generally not done consistently in the manuscript. After introducing the chemical abbreviations these could be used consequently throughout the manuscript.

P13, L13: in the WACCM -> in WACCM

P13, L33: Check sentence, looks like something went wrong here.

P16, Fig 6 caption, 4th line: Either use singular or plural, but not a mixture (an ozone mixing ratio, or ozone mixing ratios).

P18, L3: add "the" -> "in the future"

P20, Fig 8 caption, last line: Would suggest to rewrite the sentence to: Note the different y-axes for the three rows".

P23, Fig 9 caption, 1st line: add "the" -> in the 10-day …..

P23, L1: in the highest -> is the highest

P23, L7: Change "This results in a net" to "This result of the" and change "in this"  -> "at this". Generally, this sentence needs to be checked again if everything is correct.

P24, Fig. 10 caption, L2: in 10-day -> in the 10-day

P24, Fig. 10 caption, L2: in the -> for the

P24, L4: delete "for"

P25, L8: in future -> in the future

P25, L9: and specific -> and the specific

P25, L27: This are -> This is

P27, L32: I would suggest to rewrite the sentence as follows:……. increases by 60% in the subtropics (30-35°N) and by 67 % in the extra-tropics (44-49°N) by the end of …...

P28, L1: dynamic -> dynamically

P28, L7: inter annual -> inter-annual

P29, L19: in GLENS simulation -> in the GLENS simulation

P29, L21: Sentence not clear. Please check.

P30, L1: fluctuations dynamical forcing -> fluctuations of dynamical forcing